# Inferring joint sequence-structural determinants of protein functional specificity

Andrew F Neuwald[1,2]*, L Aravind[3], Stephen F Altschul[3]

[1]Institute for Genome Sciences, University of Maryland School of Medicine, Baltimore, United States; [2]Department of Biochemistry and Molecular Biology, University of Maryland School of Medicine, Baltimore, United States; [3]National Center for Biotechnology Information, National Library of Medicine, National Institutes of Health, Bethesda, United States

**Abstract** Residues responsible for allostery, cooperativity, and other subtle but functionally important interactions remain difficult to detect. To aid such detection, we employ statistical inference based on the assumption that residues distinguishing a protein subgroup from evolutionarily divergent subgroups often constitute an interacting functional network. We identify such networks with the aid of two measures of statistical significance. One measure aids identification of divergent subgroups based on distinguishing residue patterns. For each subgroup, a second measure identifies structural interactions involving pattern residues. Such interactions are derived either from atomic coordinates or from Direct Coupling Analysis scores, used as surrogates for structural distances. Applying this approach to N-acetyltransferases, P-loop GTPases, RNA helicases, synaptojanin-superfamily phosphatases and nucleases, and thymine/uracil DNA glycosylases yielded results congruent with biochemical understanding of these proteins, and also revealed striking sequence-structural features overlooked by other methods. These and similar analyses can aid the design of drugs targeting allosteric sites.
DOI: https://doi.org/10.7554/eLife.29880.001

*For correspondence:
aneuwald@som.umaryland.edu

**Competing interests:** The authors declare that no competing interests exist.

## Introduction

Residues remote from an enzyme's active site can influence catalytic activity and substrate specificity. It has been proposed that an enzyme generally has multiple conformational states that modulate its function, with residues remote from the active site often shifting the enzyme's conformational equilibrium to favor interactions associated with specific substrates or reactions (*Ramanathan et al., 2014*; *Bhabha et al., 2015*; *Whitney et al., 2016*; *Campbell et al., 2016*). Computational methods can help identify such functionally relevant non-active-site residues and their interactions. For example, direct coupling analysis (DCA) (*Morcos et al., 2011*), which predicts structural contacts from covarying residue pairs in a multiple sequence alignment (MSA), has been used to infer major conformational transitions for Hsp70 chaperones (*Malinverni et al., 2015*) and to explain the conformational heterogeneity seen in molecular dynamics simulations (*Sutto et al., 2015*). Statistical Coupling Analysis (SCA) (*Lockless and Ranganathan, 1999*) seeks to identify structural pathways of 'energetic connectivity' by applying principal component analysis to a covariance matrix to identify groups of coevolving residue positions (*Halabi et al., 2009*). SCA has been used to design proteins (*Reynolds et al., 2013*) and to predict surface sites (*Reynolds et al., 2011*) and hydrophobic cavities (*Tanwar et al., 2013*) involved in allosteric regulation. Here, we investigate residue interaction networks by combining two correlation analysis methods distinct from DCA and SCA (see Figure 7): Bayesian Partitioning with Pattern Selection (BPPS) (*Neuwald, 2014a*; *Neuwald, 2014b*), which

identifies arbitrarily large correlated residue patterns arising through evolutionary divergence, and Structurally Interacting Pattern Residues' Inferred Significance (SIPRIS), which we first describe here.

BPPS relies on the observation that protein superfamilies often diverge into subgroups, each adapting the superfamily's structural core to fill a functional niche. Often a subgroup $G$ diverges further into smaller subgroups, each conserving residues constrained by $G$'s function, as well as other residues constrained by more specialized functions. Repeated rounds of such divergence have led to hierarchically arranged subgroups, each of which conserves distinctive residues at particular positions. BPPS identifies and characterizes these subgroups by partitioning an MSA into a hierarchically nested series of MSAs, a hiMSA, based on correlated residue patterns that are distinctive of each subgroup and that often include non-active site residues.

For each subgroup of interest, the SIPRIS program takes a BPPS-defined residue pattern as input, as well as structural coordinates for a protein from that subgroup. It then identifies the statistically most significant network of pattern residues embedded within a structurally defined cluster, with a view to suggesting hypotheses for experimental investigation. Such a network is doubly significant inasmuch as BPPS identifies significant residue patterns in the absence of structural data, whereas SIPRIS defines structural clusters in the absence of sequence data. In this way, SIPRIS may statistically validate the output of BPPS or other sequence-based methods. Of course, a set of residues identified by a sequence-based method may still be biologically relevant despite a lack of SIPRIS-assigned significance. However, as we illustrate, BPPS-SIPRIS analyses often elucidate sequence/structural properties that conventional computational and experimental approaches have failed to detect.

## Results

SIPRIS takes as input: (1) structural coordinates for a protein of interest; (2) a set of residues defined by BPPS; and, optionally, (3) a predefined cluster of residues, or a starting residue defined either explicitly or as the residue closest to a 'focal point' molecule or atom. If a third input is absent, then SIPRIS uses each of the BPPS-defined residues as a starting residue, in turn, and returns the most significant result. Nested clusters are defined around a starting residue in one of three ways: (i) 'Spherical expansion', which sequentially adds residues closest to the starting residue, which thus forms the center of each cluster. (ii) 'Core expansion', which sequentially adds the residue closest to a residue within the cluster's 'core'. This core is defined as the starting residue $R$ plus all cluster residues whose distance to their $k^{th}$ closest cluster residue is less than $R$'s distance to its $k^{th}$ closest cluster residue (with $k = 7$ by default; this was selected empirically to avoid both spherical- and tentacle-shaped clusters). In this case, the cluster typically expands less symmetrically. (iii) Hydrogen-bond-network expansion, which sequentially adds a residue forming the closest sidechain-to-sidechain or sidechain-to-backbone hydrogen bond with a cluster residue. (iv) For spherical or core clustering, SIPRIS may also take DCA scores (*Marks et al., 2012*, *2011*) as a surrogate for 3D structural distances. SIPRIS evaluates the intersection between clusters and BPPS-defined residue sets with a p-value.

We applied BPPS-SIPRIS to a GCN5-like N-acetyltransferase (GNAT), several P-loop GTPases, an RNA Superfamily-II helicase, several members of the Synaptojanin/Exonuclease-Endonuclease-Phosphatase (EEP) superfamily, and two uracil/thymine DNA glycosylases. These results are summarized in *Table 1*. (Go to sipris.igs.umaryland.edu for BPPS output alignments.)

### Distinct N-acetyltransferase cofactor- and substrate-binding subdomains

GNATs catalyze the transfer of a carboxylic acyl group from Coenzyme A (acyl-CoA) to a diversity of substrates. Previously, a BPPS analysis of glucosamine-6-phosphate *N*-acetyltransferase (Gna1) led to two observations (*Neuwald and Altschul, 2016a*) (*Figure 1*): (1) Within the homodimeric structure of Gna1 (pdb: 4ag9) (*Dorfmueller et al., 2012*), BPPS-defined residues for this family are contributed by both subunits to form the dimeric interface and the active site for each subunit. In contrast, within a single subunit most of these residues are far from the active site and face away from it. Thus, the BPPS analysis implicates family-specific residues in the formation of this unusual substrate-binding pocket between subunits. (2) Residues conserved in the GNAT superfamily cluster within an acyl-CoA binding subdomain distinct from the homodimer/substrate interacting

**Table 1.** Summary of BPPS-SIPRIS results for the most significant cluster in each test case.

| Protein | PDB Structure | SIPRIS mode[*] | Focal point[†] | BPPS-SIPRIS[‡] Dist. | Init. | Term. | SIPRIS p-value | Tree level[§] | Interpretive comments[#] |
|---|---|---|---|---|---|---|---|---|---|
| Gna1 | 4ag9A | p=BDF | - | 22 | 57 | 71 | $8.5 \times 10^{-7}$ | 1 | Substrate and homodimeric interfaces |
| | | S | CoA | 17 | 41 | 87 | $6.8 \times 10^{-5}$ | 0 | CoA-binding subdomain |
| | | S | - | 23 | 56 | 72 | $9.3 \times 10^{-6}$ | 1 | DCA-based clustering |
| | | S | - | 14 | 21 | 107 | $2.5 \times 10^{-4}$ | 1 | Structure-based clustering |
| Rho1 | 3refB | B | - | 20 | 53 | 100 | $8.3 \times 10^{-5}$ | 1 | (Active site secondary shell) |
| | | C | - | 22 | 55 | 98 | $7.8 \times 10^{-7}$ | 1 | " " " " |
| Rab4 | 1z0kA | S | - | 10 | 11 | 153 | $2.1 \times 10^{-5}$ | 1 | (Active site secondary shell) |
| | | C | - | 25 | 91 | 73 | $2.6 \times 10^{-6}$ | 1 | " " " " |
| | | p=B | - | 14 | 23 | 141 | $2.9 \times 10^{-8}$ | 2 | Interface with Rabenosyn-5 |
| | | S | - | 22 | 42 | 122 | $4.8 \times 10^{-10}$ | 2 | " " " " |
| Rab8 | 3qbtA | p=B | - | 13 | 23 | 139 | $5.2 \times 10^{-7}$ | 2 | Interface with Ocrl1 |
| | | p=B | - | 12 | 23 | 139 | $6.1 \times 10^{-6}$ | 3 | Interface with Ocrl1 helix |
| | 4lhwB | p=A | - | 10 | 14 | 148 | $8.7 \times 10^{-7}$ | 2 | Homodimeric interface |
| EF-Tu | 1ob5A | S | - | 18 | 33 | 150 | $1.4 \times 10^{-7}$ | 1 | (GTP to tRNA allosteric link) |
| | | S | - | 23 | 71 | 112 | $1.0 \times 10^{-6}$ | 2 | (GTP/tRNA allosteric link to β-barrel) |
| | | S | 1B | 22 | 81 | 102 | $1.3 \times 10^{-5}$ | 1 | Cluster around 5' base 1 of tRNA |
| | | S | 2B | 18 | 47 | 136 | $2.6 \times 10^{-6}$ | 1 | Cluster around 5' base 2 of tRNA |
| | 1efuA | S | 81B | 14 | 49 | 128 | $5.2 \times 10^{-5}$ | 1 | (Nucleotide exchange allosteric network) |
| | 4zv4A | S | 291C | 21 | 66 | 109 | 0.0060 | 1 | (Mediates hijacking by Tse6 toxin) |
| CysN | 1zunB | S | - | 23 | 79 | 118 | $6.3 \times 10^{-5}$ | 2 | (Allosteric link to β-barrel domain) |
| eIF4AIII | 3ex7H | p=J | - | 11 | 18 | 128 | $6.4 \times 10^{-6}$ | 1 | (ATP to RNA allosteric link) |
| | | S | 4J | 13 | 18 | 128 | $5.1 \times 10^{-7}$ | 1 | Cluster around RNA rotation bond |
| | | S | 5J | 16 | 41 | 105 | $5.5 \times 10^{-4}$ | 1 | " " " " " |
| APE1 | 5dfiA | H | 11P | 9 | 13 | 238 | $5.2 \times 10^{-6}$ | 0 | Abasic site H-bond network |
| | | H | 11P | 22 | 99 | 152 | $1.6 \times 10^{-6}$ | 1 | " " " " |
| | | H | - | 25 | 137 | 114 | $1.7 \times 10^{-6}$ | 1 | (Active site secondary shell) |
| | | H | 9P | 25 | 137 | 114 | $1.9 \times 10^{-7}$ | 1 | H-bond network positioning abasic site |
| | | H | 12P | 23 | 119 | 132 | $7.6 \times 10^{-6}$ | 1 | " " " " " |
| Inpp5b | 4cmlA | S | - | 24 | 69 | 216 | $5.8 \times 10^{-13}$ | 0 | Active site core residues |
| | | S | - | 21 | 77 | 208 | $3.9 \times 10^{-7}$ | 1 | (Substrate recognition with allosteric link) |
| | | S | - | 12 | 30 | 255 | 0.0022 | 2 | (Membrane substrate sequestration) |
| Inpp5b | 3mtcA | S | - | 22 | 91 | 194 | $8.0 \times 10^{-7}$ | 1 | (Substrate recognition with allosteric link) |
| | | S | - | 12 | 29 | 256 | 0.0015 | 2 | (Membrane substrate sequestration) |
| Inpp5e | 2xswA | S | - | 25 | 140 | 148 | $3.7 \times 10^{-7}$ | 1 | (Substrate recognition with allosteric link) |
| | | S | - | 9 | 13 | 275 | $3.6 \times 10^{-4}$ | 2 | (Membrane substrate sequestration) |
| SHIP2 | 4a9cA | S | - | 17 | 38 | 260 | $6.0 \times 10^{-8}$ | 1 | (Substrate recognition with allosteric link) |
| | | S | - | 4 | 4 | 294 | 0.30 | 2 | (Membrane substrate sequestration) |
| TDG | 5hf7A | H | 17D | 19 | 97 | 76 | $4.1 \times 10^{-4}$ | 1 | H-bond network around excised base |
| | | H | - | 20 | 98 | 75 | $3.5 \times 10^{-5}$ | 1 | H-bond network around catalytic water |
| UDG | 2dp6A | B | - | 13 | 17 | 121 | $1.7 \times 10^{-5}$ | 1 | H-bond network distinct from TDG |

[*]Modes: S, spherical expansion; C, core expansion; H, hydrogen bond expansion (involving sidechain interactions); B, hydrogen bond expansion (also involving backbone-to-backbone interactions); P, predefined clustering (residues in the cluster are those interacting with the chain(s) whose pdb identifiers are given to the right of the equal sign).

[†]Focal points defining starting residue(s): '-',analysis was optimized over multiple starting residues (i.e., no focal point); *CoA*, cluster initiated from the residue closest to Coenzyme A; *others*, cluster initiated from the residue closest to the indicated position and chain (e.g., 1B = position 1 in pdb chain B).

‡Nature of the optimum cluster: dist., the number of distinguishing residues within the cluster (total = 25); init., the total number of residues within the cluster; term., the number of residues outside of the cluster.

§Codes designate pattern residue class: 0, superfamily; 1, family; 2, subfamily; 3, sub-subfamily. In the figures, these correspond to residues with yellow, red, orange and green sidechains, respectively.

#Comments in parentheses indicate possible functions.

DOI: https://doi.org/10.7554/eLife.29880.002

subdomain. This raises the question: How likely is such a structural distribution of these family and superfamily residues to have occurred by chance?

SIPRIS returns a p-value of $8.5 \times 10^{-7}$ for the intersection between Gna1-family residues and the predefined cluster of 57 residues contacting either the substrate or the other subunit (for residues

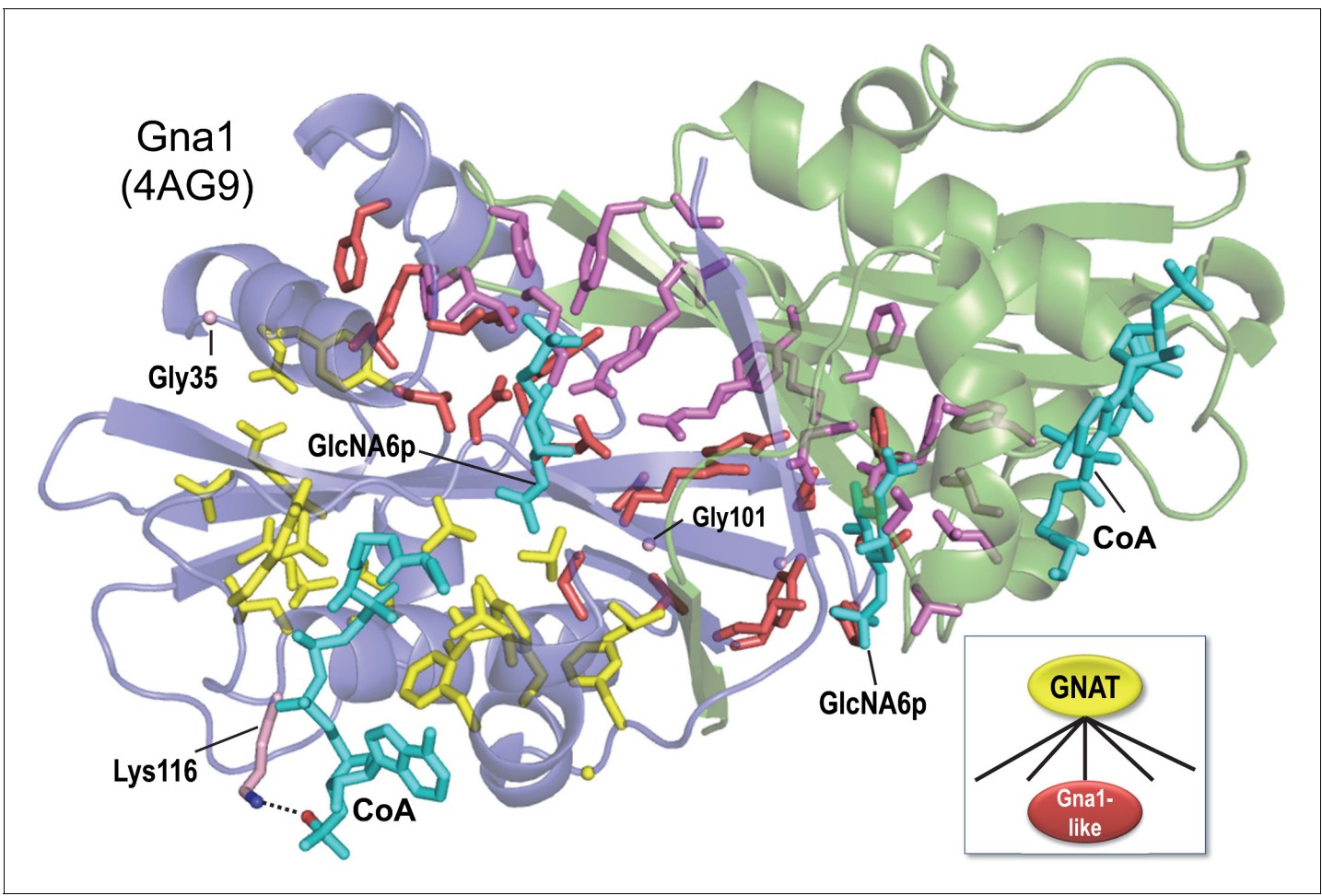

**Figure 1.** BPPS-SIPRIS analysis of the GNAT superfamily and Gna1-family based on structural coordinates for Gna1 (pdb: 4ag9) (*Dorfmueller et al., 2012*). SIPRIS clearly associates Gna1-residues with the substrate and homodimeric interfaces (p=$8.5 \times 10^{-7}$). Color scheme: homodimer subunits A and B, green and blue backbones, respectively; BPPS-defined Gna1-family residues in subunits A and B, magenta and red sidechains, respectively (glycine residues are shown as $C_\alpha$ atom spheres); GNAT superfamily residues, yellow sidechains; ligands, cyan. Lys116 (shown in light red) is outside of the SIPRIS defined cluster, but forms a hydrogen bond to a CoA phosphate group. BPPS-SIPRIS spherical clustering identified the GNAT superfamily residues shown (p=$1.7 \times 10^{-5}$). The following figure supplement and source data are available for *Figure 1*.

DOI: https://doi.org/10.7554/eLife.29880.003

The following source data and figure supplement are available for figure 1:

**Source data 1.** Contrast alignments for Gna1 N-acetyltransferase.

DOI: https://doi.org/10.7554/eLife.29880.005

**Figure supplement 1.** Applying SIPRIS to the Gna1 protein in conjunction with various methods.

DOI: https://doi.org/10.7554/eLife.29880.004

conserved across GNATs, the corresponding p-value was 0.96). Among the 25 Gna1-family residues defined by BPPS, 22 intersect with the structurally defined cluster. The three remaining residues may perform complementary functions: Gly35 and Gly101 by imparting backbone flexibility and Lys116 by helping properly position CoA via interaction with a CoA phosphate group.

SIPRIS returns a p-value of $6.8 \times 10^{-5}$ for the intersection between a (spherical) CoA-centered cluster and the set of residues conserved in all GNATs. (The corresponding p-value for Gna1-family residues is >0.99.) Of the 25 residues most distinctive of GNATs, 17 are among the 41 residues of this CoA-centered cluster. Hence, in the absence of explicit structural information, BPPS detects structurally and presumably biologically relevant features: GNAT-residues that map to an acetyl-CoA-binding module and Gna1-family residues that map to a substrate-specific 'reaction chamber' facilitating acetylation of glucosamine-6-phosphate.

## DCA-based SIPRIS analysis

Spherical clustering using residue-to-residue pseudo-distances based on DCA pairwise scores (instead of actual structural distances) likewise identifies these Gna1 structural features. In fact, the DCA-based p-value for Gna1-family residues ($9.3 \times 10^{-6}$) was more significant than the corresponding structurally based p-value ($2.5 \times 10^{-4}$). We suggest two possible reasons for this. First, DCA scores are based on multiple sequences (1200 in this case) and thus implicitly on multiple structures rather than one. Second, DCA scores should be affected by pairwise contacts between homodimeric subunits, whereas SIPRIS currently considers distances only within a single subunit. Thus, DCA- and structurally-based analyses provide somewhat different perspectives.

## Likely determinants of GTPase family and subfamily functional specificity

P-loop GTPases, upon binding to GTP versus GDP, undergo a conformational change in their so-called switch I and switch II regions that depends on the presence of a γ-phosphate group; this acts as a signal to downstream cellular components. We applied SIPRIS to two major subgroups: Rab/Rho/Ras/Ran GTPases (termed R$^4$) and translation factor (TF) GTPases (*Figure 2A*).

R$^4$ GTPases function as on/off switches regulating cellular processes. GTPase activating proteins (GAPs) facilitate hydrolysis of bound GTP (the 'on' state) to GDP (the 'off' state). Guanine nucleotide exchange factors (GEFs) turn GTPases back on by stimulating replacement of GDP with GTP. SIPRIS identifies a significant network of BPPS-defined R$^4$ residues. In Rho1 GTPases, this appears within a hydrogen-bond cluster (p=$8.3 \times 10^{-5}$; *Figure 2B*) or within a core cluster (p=$7.8 \times 10^{-7}$). In most Rab GTPases, this network often appears within a spherical or core cluster (e.g., *Figure 2C*) and, rarely, within a hydrogen-bond cluster (e.g. Rab9, pdb:1s8f [*Wittmann and Rudolph, 2004*]; p=$9.0 \times 10^{-4}$). We postulate that a significant hydrogen-bond network forms only in certain conformations. These R$^4$ sequence/structural configurations correspond to features identified through previous analyses, including: (i) Several aromatic-CH-π interactions proposed to stabilize β-strands (*Merkel and Regan, 1998*) associated with the P-loop and with the guanine binding loop, and to facilitate guanine nucleotide exchange (*Neuwald, 2009a*) (Phe99-Gly131 and Trp114-Gly27 in *Figure 2B*). (ii) A salt bridge also associated with the guanine-binding loop (Arg137-Glu163 in *Figure 2B*). (iii) Residues forming a switch II 'charge dipole pocket' proposed to facilitate conformational changes associated with the switching mechanism (*Neuwald, 2009b*). And (iv) glutamine and glutamate residues proposed to function in GTP hydrolysis (*Vetter and Wittinghofer, 2001*) and nucleotide exchange (*Gasper et al., 2008*), respectively. We propose that, together, these residues, which adjoin the GTP-binding site from the guanine-binding loop to the γ-phosphate interacting switch II region, constitute in large part the R$^4$ switching mechanism.

SIPRIS identifies a network of residues distinctive of the Rab subfamily of R$^4$ GTPases within a spherical cluster in the switch I and II regions (p=$4.8 \times 10^{-10}$ for Rab4). Rab subfamily residues also intersect with those residues contacting Rab-binding domains, with high significance based on pre-defined clustering: for Rab4-Rabenosyn-5 (*Figure 2C*) (*Eathiraj et al., 2005*) and Rab8a-Ocr1 (*Hou et al., 2011*) (*Figure 2D*) p=$2.9 \times 10^{-8}$ and $5.2 \times 10^{-7}$, respectively. This occurs despite the Rabenosyn and Ocrl1 domains being structurally distinct. Rab subfamily residues are similarly enriched at the Rab8a homodimeric interface (p=$8.7 \times 10^{-7}$) (*Figure 2E*) (*Guo et al., 2013*), supporting the notion that these residues can interact with diverse structural folds. For the Rab4

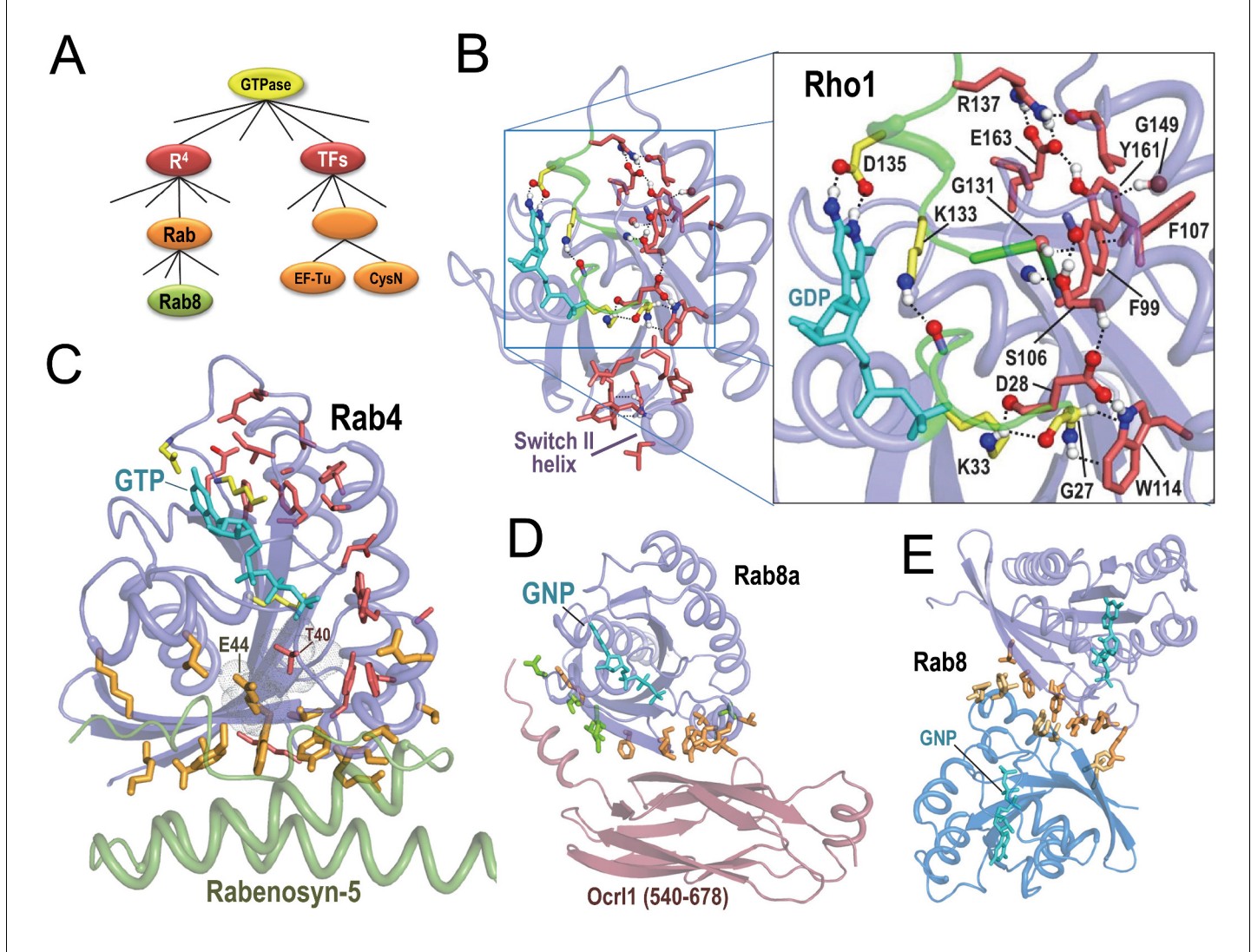

**Figure 2.** BPPS-SIPRIS analysis of R[4] P-loop GTPases. Bound guanine nucleotide (shown in cyan) allows orientation of each subfigure relative to the others. (**A**). BPPS-defined hierarchical relationships among the GTPases examined here. (**B**). *Entamoeba histolytica* Rho1 GTPase (pdb: 3refB) (**Bosch et al., 2011**). Color scheme: R[4]-specific residues forming a BPPS-SIPRIS-defined hydrogen-bond network (p=8.3 × 10$^{-5}$), red sidechains; residues conserved in P-loop GTPases and interacting with bound guanine nucleotide, yellow sidechains; atoms forming hydrogen bonds, CPK coloring. Modeled hydrogen atoms were generated using the Reduce program (**Word et al., 1999**). (**C**). Rab4 bound to GTP and to the Rab-binding domain of Rabenosyn (pdb: 1z0kA [**Eathiraj et al., 2005**]). BPPS-SIPRIS-defined residues distinctive of R[4] (red sidechains) and Rab (orange) have core and Rabenosyn-contacting predefined cluster p-values of 2.6 × 10$^{-6}$ and 2.9 × 10$^{-8}$, respectively. The sensor threonine (Thr40) has substantial van der Waals contact with Glu44; Thr40 is a R[4]-specific (red) residue outside of the SIPRIS-defined cluster. (**D**). Rab8a in complex with the GTP analog, GNP, and with Ocrl1 (residue 540–678) (pdb: 3qbtA) (**Hou et al., 2011**]). Residues distinctive of Rab GTPases (orange) and of the Rab8 subgroup (green) are enriched at the Ocr1 interface (p=5.2 × 10$^{-7}$ and 6.1 × 10$^{-6}$, respectively). (**E**). Rab8a homodimeric complex (pdb: 4lhwAB) (**Guo et al., 2013**). Rab-specific residues (orange) are enriched at the homodimeric interface (p=8.7 × 10$^{-7}$). The following source data are available for *Figure 2*.
DOI: https://doi.org/10.7554/eLife.29880.006

The following source data is available for figure 2:

**Source data 1.** Contrast alignments for Rab8, Rab4 and Rho1 GTPases.
DOI: https://doi.org/10.7554/eLife.29880.007

structure in *Figure 2C*, Thr40, another R[4]-specific residue, albeit one outside of the SIPRIS-defined cluster, corresponds to the switch I residue that senses the γ-phosphate of GTP. This residue establishes its greatest contact area (45 Å$^2$) with Glu44, one of the Rab-specific residues contacting Rabenosyn-5; thus Thr40 and Glu44 may link sensing of the γ-phosphate to substrate binding. For Rab8a

both Rab- and Rab8-specific residues appear to mediate binding to the Ocr1 domain (*Figure 2D*); in all, 19 of the 23 Rab8-Ocrl1 interface residues are distinctive of either the Rab subfamily or the Rab8 sub-subfamily. Many of the Rab8-residues interact with an N-terminal helix extending out of the Ocrl1 β-sandwich domain, perhaps thereby compensating for the lack of binding specificity of Rab-subfamily residues.

BPPS grouped translation factor (TF) GTPases into a single family (*Figure 2A*), which includes initiation factors (e.g. IF2 and eIF5B), sulfate adenyltransferases (CysN), ribosome-releasing factor 2, peptide chain release factor 3, elongation factors EF-Tu, EF1α and selenocysteine-specific elongation factor, EF4, aEF2, and EF-G (*Leipe et al., 2002*). Within *Thermus aquaticus* EF-Tu complexed with a GTP analog, Phe-tRNA, and the antibiotic Enacyloxin IIA (*Parmeggiani et al., 2006*), TF-specific residues (*Figure 3A*) spherically cluster around the switch I and II and P-loop regions ($p = 1.4 \times 10^{-7}$); this differs from the $R^4$-residue arrangement in *Figure 2B*. The two 5'-terminal tRNA nucleotide bases, which base-pair with the 3' strand to which the aminoacyl group is attached, establish the greatest contact with the EF-Tu GTPase domain among all the bases of the tRNA. TF-specific residues cluster around these 5' bases ($p = 1.3 \times 10^{-5}$ and $2.6 \times 10^{-6}$, respectively) and link the 5' region of aa-tRNA to the GTP γ-phosphate; this cluster includes Thr62, which senses γ-phosphate. We hypothesize that, upon correct tRNA-anticodon pairing with its mRNA codon, these TF residues assist in coupling GTP hydrolysis to coordinated conformational changes that dissociate EF-Tu from the ribosome and from tRNA, which can then fully enter the ribosomal A site.

TF-specific residues also may be important for guanine nucleotide exchange mediated by EF-Ts. Within the structure of EF-Tu bound to EF-Ts (pdb: 1efu) (*Kawashima et al., 1996*), 14 TF-residues form a (spherical) cluster ($p = 5.2 \times 10^{-5}$; *Figure 3B*) centered on Phe81 of EF-Ts, the residue with the greatest area of contact with EF-Tu. These TF-residues, which include His19, His84, and Gln114 of EF-Tu, adjoin two regions of EF-Ts contacting EF-Tu and are conserved across bacteria and eukaryotes (*Figure 3B*). His19, which is located in the P-loop of EF-Tu, is the residue that is most characteristic of these translation factors. Both His19 and Gln114 have been implicated in nucleotide exchange (*Zhang et al., 1998*), and in destabilization of $Mg^{+2}$ coordination (leading to guanine nucleotide release) upon intrusion of EF-Ts Phe81 near His84 of EF-Tu (*Schümmer et al., 2007*). Given recent evidence for an EF-Tu/Ts·GTP·aa-tRNA quaternary complex (*Burnett et al., 2014*), we conjecture that TF-residues may help couple GTP-hydrolysis-mediated loading of aa-tRNA onto the ribosome with nucleotide exchange by EF-Ts. *P. aeruginosa* Tse6 toxin (*Whitney et al., 2015*) appears to have hijacked this TF interaction interface with EF-Ts (*Figure 3C*).

BPPS partitions EF-Tu and CysN into a common subfamily within the TF family, consistent with earlier analysis supporting their specific relationship (*Leipe et al., 2002*; *Inagaki et al., 2002*). CysN, together with the catalytic CysD subunit, form a sulfate adenylyltransferase complex involved in sulfur assimilation. The CysND-catalyzed reaction is analogous to the first step in charging a tRNA, and CysN's contact sites with CysD are similar to, and include residues homologous to, EF-Tu's contact sites with aa-tRNA. Within the CysND complex (pdb: 1zun) (*Mougous et al., 2006*) EF-Tu/CysN-residues cluster around the switch I and II regions ($p = 6.3 \times 10^{-5}$; *Figure 3D*). In CysN, these residues adjoin contact regions with CysD and with the CysN C-terminal linker and β-barrel domains. Analogously in EF-Tu, they are proximal to the contact region with aa-tRNA and the EF-Tu C-terminal linker and β-barrel domains (*Figure 3E*). Within EF-Tu these residues are also located between the bound antibiotic enacyloxin IIA and the GTPase- and TF-specific residues (*Figure 3A*). Because enacyloxin IIA hinders the release of EF-Tu-GDP from the ribosome (*Parmeggiani et al., 2006*), we hypothesize that these residues may help mediate this process.

## Comparison of two P-loop NTPase superfamilies: eIF4AIII RNA helicase

For comparison, we analyzed another nucleic-acid-associated P-loop NTPase, the Superfamily II RNA helicase eIF4AIII, which is a component of the exon junction complex (EJC). The EJC deposits onto spliced mRNAs and plays an important role in mRNA transport, translation, and quality control. RNA helicases are part of a huge group of NTPases that undergo ATP-hydrolysis-coupled conformational changes to unwind double-stranded nucleic acids, translocate nucleic acids or re-distribute protein complexes on nucleic acids (*Anantharaman et al., 2002*; *Bourgeois et al., 2016*; *Lohman et al., 2008*; *Northall et al., 2016*). For the transition state structure of eIF4AIII bound to RNA, a predefined cluster of RNA helicase-specific residues contacting RNA is highly significant ($p = 6.4 \times 10^{-6}$; *Figure 3F*). Focal point spherical clustering indicates that these residues are centered on RNA bases

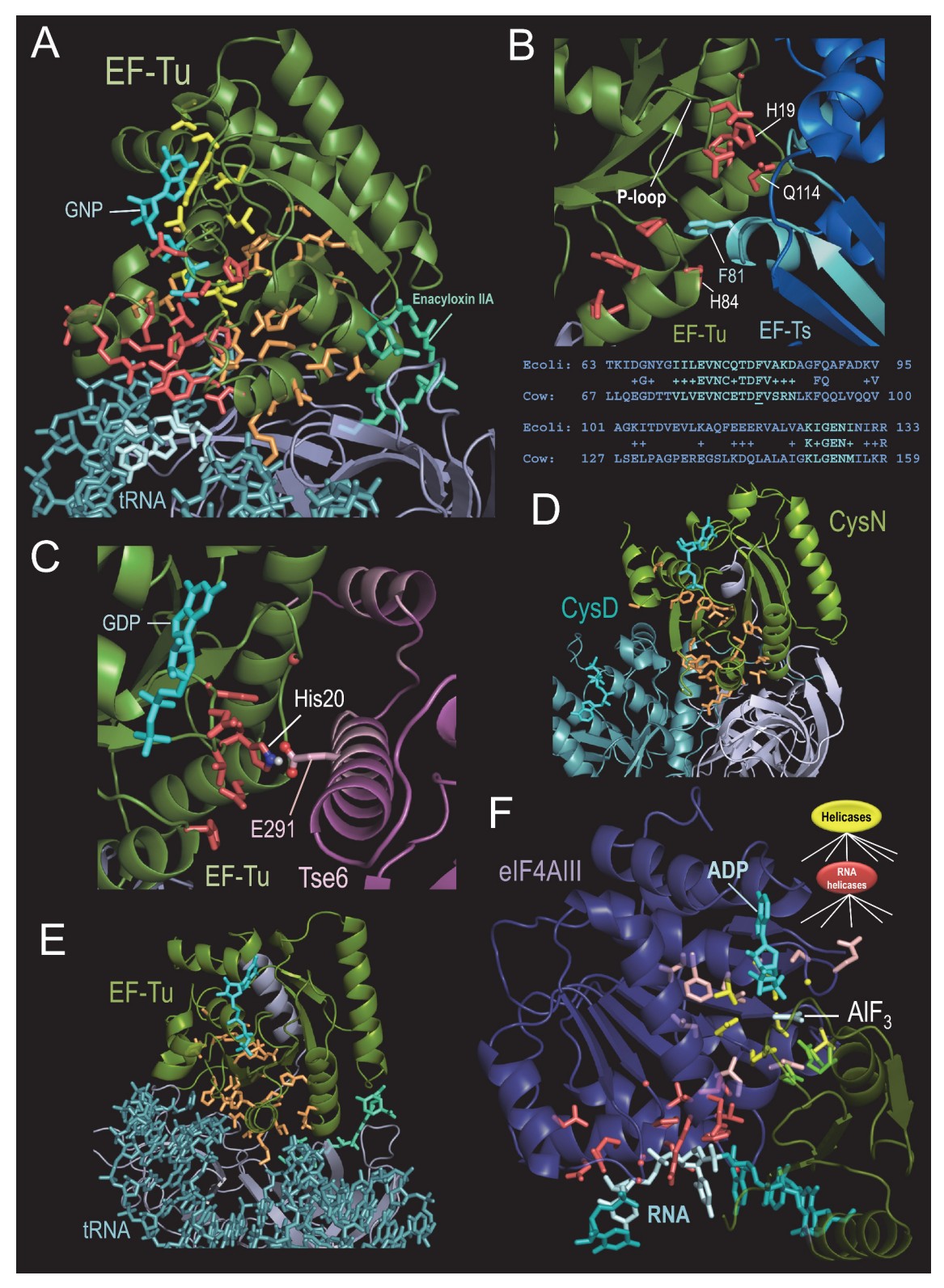

**Figure 3.** BPPS-SIPRIS analysis of translation-associated P-loop NTPases. (**A**). *Thermus aquaticus* EF-Tu complexed with the antibiotic enacyloxin IIA, a GTP analog, and Phe-tRNA (pdb: 1ob5) (*Parmeggiani et al., 2006*). Color scheme: BPPS-SIPRIS defined GTPase-, TF- and EF-Tu/CysN-specific residues, yellow, red, and orange sidechains, respectively; GTPase domain backbone, green; C-terminal β-barrel domains, gray; phe-tRNA, teal; 5' end nucleotide bases, light cyan; guanine nucleotide, cyan; enacyloxin IIA, greenish-cyan. Spheres indicate glycine Cα atoms. (**B**). BPPS-SIPRIS cluster of EF-

*Figure 3 continued on next page*

*Figure 3 continued*

Tu TF-residues centered on EF-Ts Phe81 at the EF-Tu/EF Ts interface (pdb: 1efu) (*Kawashima et al., 1996*). Regions in EF-Ts conserved between *E. coli* and cow are shown in cyan both in the figure and in the corresponding alignment below it. (C). *P. aeruginosa* EF-Tu bound to the Tse6 toxin domain (pdb: 4zv4) (*Whitney et al., 2015*). EF-Tu His20, which corresponds to His19 in (B), appears to form a salt bridge with Glu291 of Tse6. In light pink are regions of Tse6 contacting EF-Tu. Spherically clustered residues (p=0.0060) centered on Glu291 of Tse6 are shown with red sidechains. (D). Spherically clustered EF-Tu/CysN residues (orange; p=$6.3 \times 10^{-5}$) within the CysND complex (pdb: 1zun) (*Mougous et al., 2006*). (E). Spherically clustered EF-Tu/CysN-residues in EF-Tu (pdb: 1ob5) (p=$1.0 \times 10^{-6}$). (F). Human eIF4AIII bound to RNA, ADP, and the γ-phosphate transition state mimic AlF₃ (pdb: 3e × 7) (*Nielsen et al., 2009*). Color scheme: eIF4AIII N- and C-terminal domains, violet and green, respectively; RNA and ADP, cyan; AlF₃, light cyan; superfamily-conserved catalytic residues, yellow sidechains; RNA helicase-specific residues clustered on (light cyan-colored) RNA bases 4–5, red; other RNA helicase-specific residues, light red; C-terminal catalytic residues, bright green. The following source data are available for *Figure 3*.
DOI: https://doi.org/10.7554/eLife.29880.008

The following source data is available for figure 3:

**Source data 1.** Contrast alignments EFTu GTPase and eIF4AIII RNA helicase.
DOI: https://doi.org/10.7554/eLife.29880.009

---

4 and 5 (p=$5.1 \times 10^{-7}$ and p=$5.5 \times 10^{-4}$, respectively), which establish the greatest contact with the ATPase domain. These observations and a rotated bond between bases 4 and 5 suggest that these residues help couple ATP hydrolysis to disruption of duplex RNA. Clusters centered on other bases are not significant (p>0.9). Most of the remaining RNA helicase-specific residues surround key active site residues or interact with C-terminal domain catalytic residues, including two arginine fingers (*Figure 3F*). Given this configuration, ATP hydrolysis seems likely to shift the relative orientations of the N- and C-terminal domains, both of which interact with RNA.

## Residue networks adapting the EEP catalytic core to diverse substrates

EEP enzymes cleave phosphodiester bonds in substrates that include nucleic acids and phospholipids. To identify residues likely responsible for EEP functional divergence, we applied BPPS-SIPRIS to APE1, an exonuclease III-like apurinic/apyrimidinic endonuclease (exoIII-AP-endo), and several inositol polyphosphate 5-phosphatases (INPP5) (*Figure 4A*).

APE1 participates in the DNA excision repair pathway by incising the apurinic/apyrimidinic (AP) site phosphodiester backbone; this generates a single nucleotide DNA gap with 3'-hydroxyl and 5'-deoxyribose phosphate termini—a cytotoxic intermediate substrate that is then processed by DNA polymerase β (*Liu et al., 2007*). A proposed mechanism for APE1 (*Mol et al., 2000*) involves superfamily-conserved active site residues forming hydrogen bonds with the oxygen atoms of the phosphate group at the abasic site. Consistent with this, SIPRIS identifies a superfamily-conserved hydrogen-bond network centered on the abasic site (p=$5.2 \times 10^{-6}$) within a structure of APE1 bound to DNA harboring an abasic site phosphate group analog (phosphorothioate) in one strand (*Figure 4B,C*). Centering on adjacent bases in the same strand was less significant (p>0.003). For exoIII-AP-endo-conserved residues SIPRIS identifies a significant hydrogen-bond network centered on the abasic site (p=$1.6 \times 10^{-6}$) or on adjacent bases 8–9 and 12–13 (p=$1.9 \times 10^{-7}$ to $7.6 \times 10^{-6}$); these residues may contextually position catalytic residues around the abasic site. In particular, regions associated with these residues insert into the DNA major and minor grooves on either side of the abasic site, and form a kink in and engulf the target DNA strand (*Figure 4B*). Thus, exoIII-AP-endo residues appear to form a substrate-specific 'reaction chamber', as might be expected. They also tend to aggregate between the catalytic core and a loop containing basic residues that interact with the major groove of DNA (*Figure 4B*). Modification by nitric oxide (nitrosation) of one of these residues, Cys310, results in dissociation of APE1 from DNA and relocation to the cytoplasm (*Qu et al., 2007*); thus, the associated hydrogen-bond network may communicate the nitrosation signal to the DNA-binding site.

BPPS-SIPRIS-defined INPP5-residues also form a significant hydrogen bond network (p=$1.1 \times 10^{-7}$) adjacent to the superfamily-conserved cluster (*Figure 5A,B*). We hypothesize that this network recognizes inositol polyphosphates harboring phosphate groups at positions 4 and 5 of the inositol ring. INPP5 phosphatases cleave the 5-phosphate, but require for recognition the 4-phosphate, which directly interacts with three network-associated basic residues—perhaps thereby mediating substrate recognition (*Figure 5C*). In some structures, the INPP5 network residues most remote from the catalytic core are part of a cleft accommodating a phosphate or a glycerol

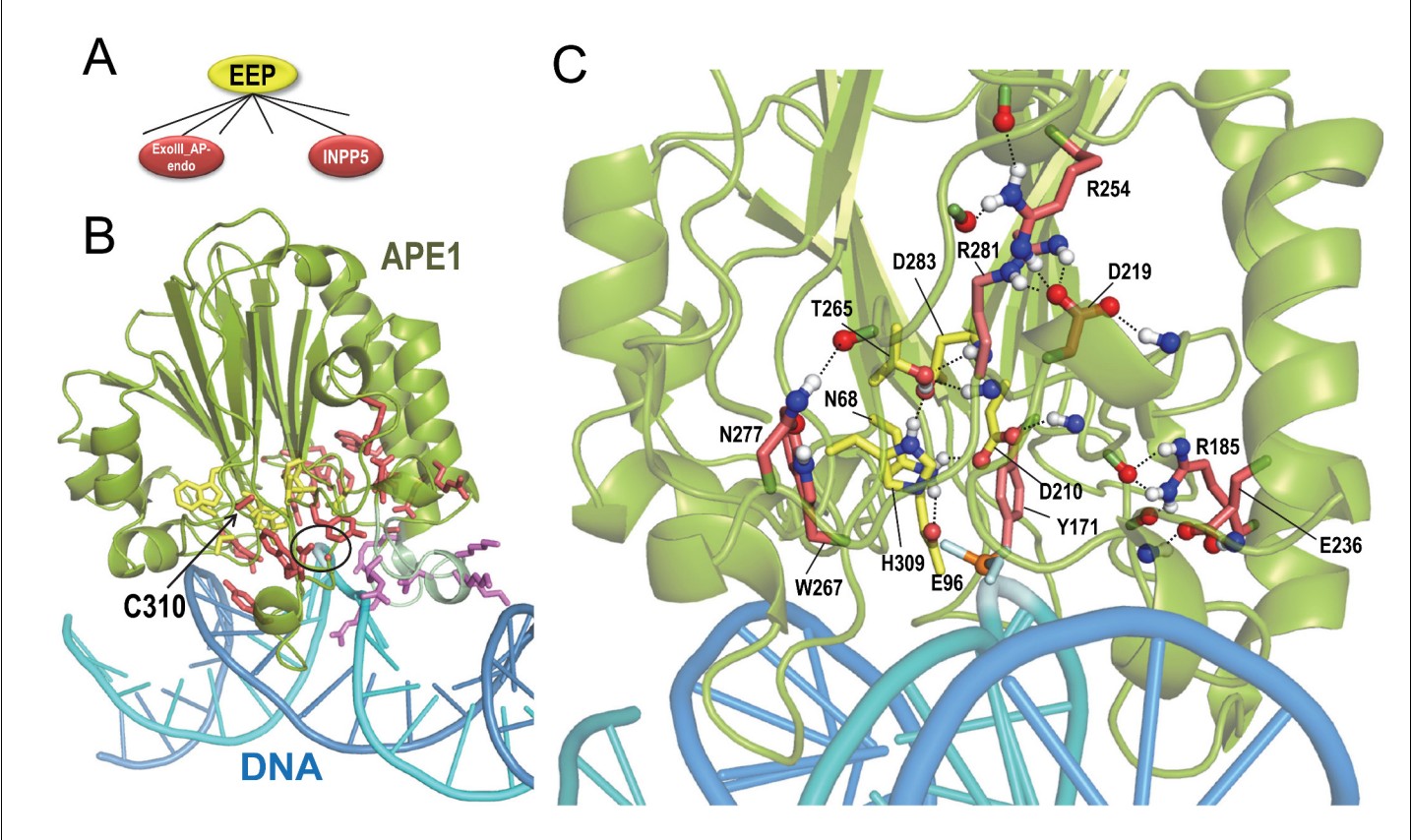

**Figure 4.** BPPS-SIPRIS analysis of synaptojanin/EEP domains. (**A**). The two major groups of the BPPS-defined EEP hierarchy examined here. (**B**). Human APE1 phosphorothioate substrate complex (pdb: 5dfi) (*Freudenthal et al., 2015*). Replacement of the phosphodiester bond with phosphorothioate prohibits cleavage by APE1 at the abasic site (circled). Cys310, which is nitrosated, is indicated. Color scheme: APE1 backbone trace, green; DNA strand containing the abasic site, cyan; complementary strand, marine blue; the BPPS-SIPRIS-defined residues distinctive of the EEP superfamily and of the exoIII-AP-endo family, yellow and red sidechains, respectively; basic residues within a loop interacting with the major groove of DNA, purple. (**C**). Close up of the APE1 active site. EEP-specific residues forming a hydrogen-bond network are shown with yellow sidechains. For clarity, only a few of the EEP- and exoIII-AP-endo-specific residues in the network are shown. The following source data are available for *Figure 4*.
DOI: https://doi.org/10.7554/eLife.29880.010

The following source data is available for figure 4:

**Source data 1.** Contrast alignments for APE1 endonuclease.
DOI: https://doi.org/10.7554/eLife.29880.011

(*Figure 5D,E*), suggesting that these may form another (unknown) membrane interaction site or an allosteric site that binds a molecule similar to the known substrate.

INPP5 proteins regulate diverse cellular processes, including postsynaptic vesicular trafficking, insulin signaling, cell growth and survival, and endocytosis. With this in mind, we examined three INPP5 subfamilies: INPP5B, INPP5E and SHIP2 (*Figure 5F*). Residues that most distinguish the INPP5B subfamily form a cluster between the proposed membrane interacting region (*Trésaugues et al., 2014*) and the EEP catalytic core (*Figure 5A*). INPP5E- and SHIP2-specific residues also cluster in this same region (*Figure 5G,H*)—although the SHIP2 cluster is not statistically significant. This suggests a possible role for these residues in sequestering specific membrane-associated phosphoinositide substrates from the lipid bilayer.

## Family-specific catalysis: thymine DNA glycosylases

Uracil DNA glycosylases (UDGs) remove uracil from DNA, thereby initiating the DNA base excision repair pathway (*Aravind and Koonin, 2000*). Uracil may be incorporated into DNA by DNA polymerase or by cytosine deamination. Thymine DNA glycosylases (TDGs) initiate base excision repair

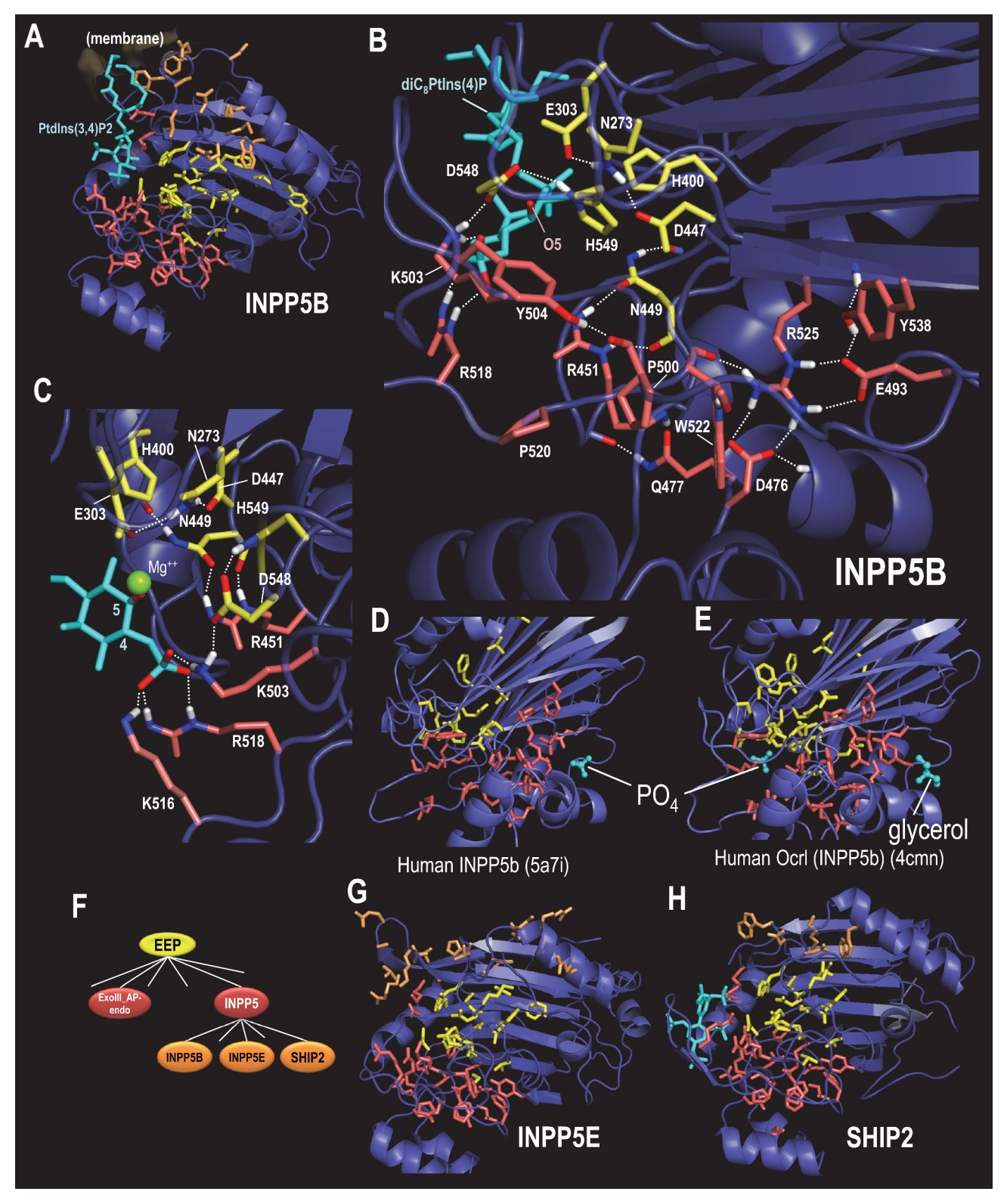

**Figure 5.** BPPS-SIPRIS analysis of synaptojanin/EEP domains within INPP5 proteins.  Color code: EEP-residues, yellow sidechains; INPP5 residues, red sidechains; INPP5B-, INPP5E- and SHIP2-subfamily residues, orange sidechains; ligands, cyan; atoms involved in hydrogen bonds, CPK coloring. (**A**). Human INPP5B in complex with phosphatidylinositol 3,4-bisphosphate (pdb: 4cml) (*Trésaugues et al., 2014*), which is associated with cytosolic and mitochondrial membranes (*Speed et al., 1995*). BPPS-SIPRIS results: EEP spherical cluster, $p=5.8 \times 10^{-13}$; INPP5 spherical cluster, $p=3.9 \times 10^{-7}$;
*Figure 5 continued on next page*

*Figure 5 continued*

INPP5B spherical cluster, p=0.0021. (B). INPP5 hydrogen bond network within human INPP5B (pdb: 3mtc) (unpublished). (C). View of INPP5-residues (in 3mtc) that bind the 4-phosphate group required for substrate recognition. (D). Human INPP5B with phosphate bound to a possible membrane interaction or allosteric site (*Mills et al., 2016*). (E). Human INPP5B Ocrl with glycerol bound to the same site as indicated in (D) (*Trésaugues et al., 2014*). (F). INPP5 subgroups within the BPPS-defined hierarchy. (G). Human INPP5E (pdb: 2xsw) (unpublished), which is associated with the primary cilium, an organelle involved in signal transduction (*Jacoby et al., 2009*) (spherical cluster, p=$3.6 \times 10^{-4}$). (H). Human SHIP2 (pdb: 4a9c) (*Mills et al., 2012*), which is associated with membrane ruffle formation (*Hasegawa et al., 2011*) (spherical cluster, p=0.30). The following source data are available for *Figure 5*.

DOI: https://doi.org/10.7554/eLife.29880.012

The following source data is available for figure 5:

**Source data 1.** Contrast alignments for INPP5 phosphatases.

DOI: https://doi.org/10.7554/eLife.29880.013

by removing T from G·T mispairs, which can be due to deamination of 5-methylcytosine. These enzymes also remove oxidized derivatives of methyl cytosine such as 5-formyl and 5-carboxymethyl cytosine, which are epigenetic marks or intermediates in the reset of 5mC marks by the TET enzymes (*Pastor et al., 2013*). Within the structure for human TDG (*Pidugu et al., 2016*) BPPS-SIPRIS identifies a significant hydrogen-bond network associated with TDG-family residues (*Figure 6A,B*); also in this network are residues classified by BPPS to a metazoan TDG subfamily. Like APE1, network residues appear to position loops containing basic residues that, in this case, interact with both the major and minor grooves of bound DNA (*Figure 6C*). Network residues also form hydrogen bonds to DNA oxygen atoms on either side of the thymine base being excised—suggesting that they may help position the substrate for catalysis by sensing particular sequence contexts (*Figure 6B*). Near the center of this network and in contact with the targeted thymine base is the residue most distinctive of metazoan TDGs, Asn230 (*Figure 6B* and *Figure 6—source data 1*); in other TDG subfamilies, a hydrophobic residue occurs at this position. Other TDG-residues in this network encase a water molecule believed to function as a nucleophile in catalysis (*Pidugu et al., 2016*) (*Figure 6D*). Hence, for TDG, family-specific residues may play a critical catalytic role. UDG harbors a hydrogen bond network distinct from that of TDG (*Figure 6E*), indicating a mechanistic divergence.

## Applying SIPRIS with other methods

Applying SIPRIS in conjunction with various protein function determining residue (FDR) methods (*Casari et al., 1995*; *Ye et al., 2008*; *Pirovano et al., 2006*; *Kalinina et al., 2004*; *Hannenhalli and Russell, 2000*; *Livingstone and Barton, 1996*; *Mihalek et al., 2004*; *Mirny and Gelfand, 2002*; *Lichtarge et al., 1996*; *Sankararaman and Sjölander, 2008*; *Fischer et al., 2008*; *Kalinina et al., 2009*; *Janda et al., 2012*; *Janda et al., 2014*; *Marttinen et al., 2006*; *Kolesov and Mirny, 2009*; *Wilkins et al., 2012*; *Chakraborty and Chakrabarti, 2015*; *Gaucher et al., 2002*; *Xin and Radivojac, 2011*; *Capra and Singh, 2008*) is straightforward in principle. However, several factors complicate comparisons to BPPS-SIPRIS. First, a fair number of published FDR methods are no longer available as source code, executables or over the world wide web (e.g. INTREPID [*Sankararaman and Sjölander, 2008*] and MINER [*La and Livesay, 2005*]). Second, many FDR methods (e.g. GroupSim [*Capra and Singh, 2008*]) require user-provided input, such as an MSA, a phylogenetic tree, or prespecified categories with corresponding sequence assignments for each category. This confounds the comparison because the contribution of each user-provided component to overall performance is unclear. In contrast, BPPS-SIPRIS requires no input beyond the query and database sequences, and its algorithmic components are statistically coherent. Third, those FDR methods not requiring user-generated input typically are based on a phylogenetic tree; this renders infeasible their application to large sequence sets, which is a key aspect of SIPRIS's ability to detect biologically relevant features. Our attempts to input even moderately large sequence sets to various FDR programs resulted in runtime errors. By focusing on a hierarchy of subgroups, each defined by a correlated residue pattern, BPPS eliminates the need for a phylogenetic tree, which would introduce more complexity than either is necessary or can be reliably inferred.

Finally, BPPS-SIPRIS aims to identify biologically relevant interaction networks whose functions are not necessarily known, whereas FDR methods generally try to identify residues responsible for well-characterized functions—such as catalysis or substrate recognition—that can be experimentally

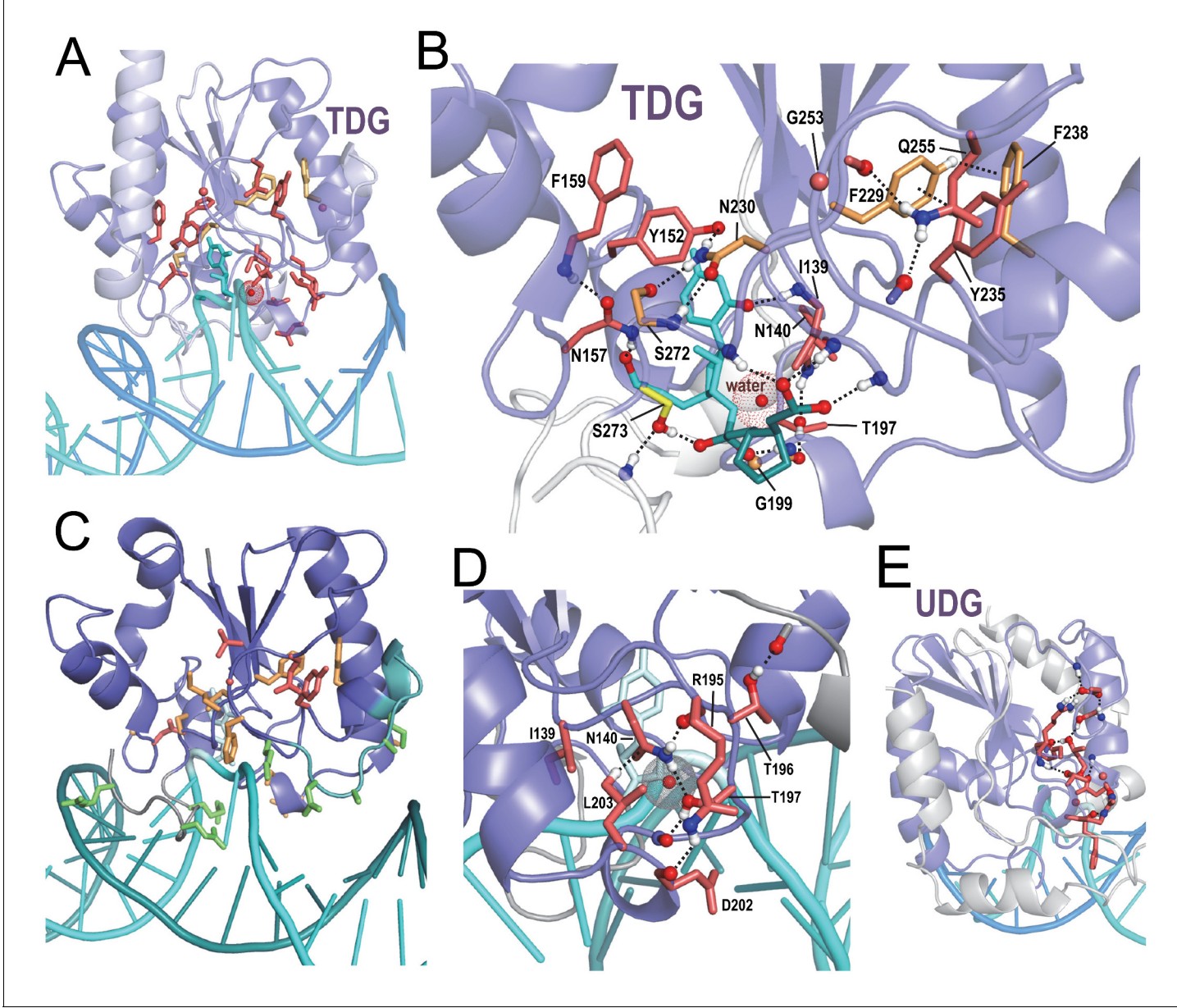

**Figure 6.** BPPS-SIPRIS analysis of DNA glycosylases. (**A**). Thymine DNA glycosylase (TDG) family (red sidechains) and metazoan subfamily (orange sidechains) residues forming a significant hydrogen bond network (p=3.5 × 10$^{-5}$) within human TDG (pdb: 5hf7) (*Pidugu et al., 2016*). (**B**). TDG H-bond network consisting of residues distinctive both of all TDGs (red sidechains) and of metazoan TDGs (orange sidechains). This network includes hydrogen bonds to DNA oxygen atoms on either side of the thymine base to be excised (cyan); note that Phe238 and Tyr235 appear to position the N-terminus of their helix to hydrogen bond to substrate backbone oxygens; another such hydrogen bond involves Ser273, a residue generally conserved in the entire superfamily. The water molecule shown may act as the nucleophile in the reaction. For clarity, not all of the BPPS-SIPRIS-defined residues are shown. (**C**). TDG hydrogen-bond network residues may help position basic residues (green sidechains) interacting with the minor and major grooves of DNA. (**D**). TDG family-specific hydrogen-bond network residues surrounding a proposed catalytic water molecule (red sphere with dot cloud). (**E**). A BPPS-SIPRIS-defined H-bond network (p=1.7 × 10$^{-5}$) distinct from that of TDG within *Thermus thermophilus* uracil DNA glycosylase (UDG) (pdb: 2dp6). The following source data are available for *Figure 6*.

DOI: https://doi.org/10.7554/eLife.29880.014

The following source data is available for figure 6:

**Source data 1.** Contrast alignments for DNA glycosylases.
DOI: https://doi.org/10.7554/eLife.29880.015

benchmarked (*Chakrabarti and Panchenko, 2009*). However, as has been noted (*Dessimoz et al., 2013*; *Jiang et al., 2014*), we lack reliable gold standards for many functionally relevant residues, due to a lack of experimental characterization. Consequently, methods designed to identify residues with specific, known functions, if successful, will tend to penalize residues involved in unknown functions. In contract, the goal of BPPS-SIPRIS is to recognize also such residues of unknown function.

With this in mind, we compared the BPPS-SIPRIS analyses in this study to SIPRIS analyses based on the FRpred (*Fischer et al., 2008*), CLIPS-1D (*Janda et al., 2012*), and Evolutionary Trace (ET) (*Lichtarge et al., 1996*; *Wilkins et al., 2012*) programs, which define residue sets given only a query sequence. These and similar methods differ from BPPS by not classifying sequences into divergent subgroups per se. Instead, FRpred seeks to classify residues as catalytic, ligand binding and subtype-specific. FRpred catalytic and ligand-binding residues generally correspond to superfamily-conserved residues, whereas FRpred subtype-specific residues fail to correspond to any BPPS subgroups. For example, when we ran the Rab4 analysis as in *Figure 2C* using FRpred-defined residue sets instead of BPPS-defined sets, the first two FRpred categories nearly entirely overlapped with each other and with the Rab4 structural core; the subtype-specific category failed to return a significant cluster (p>0.05). SIPRIS analyses of other protein domains yielded similar results. CLIPS-1D defines catalytic, ligand-binding and structural categories, which likewise fail to correspond to BPPS subgroups. ET assigns residue functional importance scores without splitting into categories, and thus fails to differentiate between BPPS subgroups. As previously noted (*Madabushi et al., 2002*), high ET-scoring residues are often clustered structurally, which SIPRIS analyses confirm. Due to methodological differences, however, BPPS-SIPRIS clustering identifies sequence/structural features distinct from these other methods, as illustrated in *Figure 1—source data 1*. Although other methods may identify biologically relevant residues different than those identified here, this study suggests that by characterizing divergent subgroups, BPPS-SIPRIS analyses can identify significant, otherwise overlooked sequence/structural properties.

## Discussion

Active site residues directly involved in catalysis are believed often to communicate with a network of other functionally important residues, some of which may be far from the active site (*Sunden et al., 2015*). The problem of identifying these networks is fundamental for understanding how proteins work. As illustrated here, BPPS-SIPRIS analyses can reveal information relevant to functional specialization by identifying statistically significant interaction networks. This includes, for example: (1) The nitrosation associated network in APE1 of the synaptojanin (EEP) superfamily. (2) The protein-protein interaction interfaces for diverse $R^4$ GTPases. (3) The protein-protein interaction interface in EF-Tu, which can be hijacked by the *P. aeruginosa* Tse6 toxin (*Whitney et al., 2015*). In each of these cases, the residue-networks identified by our analysis suggest features congruent with current biochemical understanding of these proteins. Additionally, our analyses generated the following hypotheses: (1) Family-specific residues form hydrogen bonds (*Figure 4C*) responsible for APE1 abasic site substrate specificity. (2) INPP5 family and sub-family specific residues (*Figure 5E–F*) mediate, respectively, allosteric regulation and sequestration of specific membrane-associated phosphoinositide substrates from the lipid bilayer. (3) A hydrogen bond network associated with the residue most distinctive of metazoan TDGs, Asn230 in humans, mediates substrate-specific catalysis in DNA glycosylases, perhaps related to the discrimination of epigenetic marks present in metazoan DNA (*Pastor et al., 2013*; *Zhang et al., 2012*), such as 5-fC and 5-caC.

More generally our analyses suggest: (1) Family-specific residues often form a substrate-specific 'reaction chamber' associated with the structural core and active site, as seen for Gna1-related acetyltransferases, phosphoesterases related to APE1, and DNA glycosylases. (2) Subfamily-specific residues serve subordinate roles, such as mediating interactions with effector proteins, or coupling conformational changes to signaling. In this way, the same basic structural core and catalytic mechanism may accommodate a wide variety of cellular functions.

The SIPRIS clustering strategies described here accommodate further development. For example, one might use consensus distances from multiple structures to reduce noise. An open question is the significance of multiple BPPS-SIPRIS networks for a single subgroup, analogous to that for multiple regions of similarity between two sequences (*Karlin and Altschul, 1993*). Additional strategies include: applying BPPS-SIPRIS to functionally interacting proteins, treating them as a single

sequence; and defining clusters using features such as secondary structure, surface accessibility or electrostatic potential. BPPS identifies correlated residue patterns presumably associated with functional specialization, and SIPRIS identifies correlations between defined residue sets and structural features. In contrast, DCA identifies correlations between pairs of residues that presumably interact structurally. Combining BPPS-SIPRIS with DCA may improve protein modeling and the characterization of functional interactions. Given the statistical and information theoretic foundation of these methods, one should be able to combine them in a principled manner.

In summary, the BPPS-SIPRIS system should aid the characterization of functionally interacting residues remote from protein active sites.

## Materials and methods

### BPPS-SIPRIS overview

BPPS-SIPRIS analysis involves the following steps, as illustrated in *Figure 7*: (1) MAPGAPS (*Neuwald, 2009*) detects and aligns protein database sequences containing the domain of interest starting from a representative ('seed') MSA or from an hiMSA, either of which may be either curated manually or created automatically. This generates an MSA. (2) Bayesian Partitioning with Pattern Selection (BPPS) (*Neuwald, 2014a*; *Neuwald, 2014b*; *Neuwald and Altschul, 2016a*) is applied in three steps: (i) Step 1 uses Markov chain Monte Carlo (MCMC) sampling to partition the MSA into hierarchically-arranged subgroups based on the correlated residue patterns most distinctive of each subgroup. (ii) Step 2 converts the MSA into a hiMSA based on the BPPS hierarchy. (iii) Step 3 creates subgroup 'contrast alignments' and corresponding SIPRIS input files. (3) The SIPRIS program performs pattern residue cluster analyses and, as a runtime option, will create corresponding PyMOL (*Schrodinger, 2010*) scripts for viewing clusters within 3D structures (as in *Figures 1–6*). Each step in this process applies statistical criteria to ensure significance (see below).

### Software and data availability

BPPS-SIPRIS software, source code, instructions, and the input data required to perform the analyses described here are available at sipris.igs.umaryland.edu; this includes: (1) the MAPGAPS, BPPS, and SIPRIS programs; (2) MSA format conversion programs; (3) a phylum annotation program (fatax); and (4) the full multiple sequence alignments and pdb structural coordinate files used as input to BPPS and SIPRIS. The source code is available at sourceforge (sourceforge.net/p/bpps-sipris/code/ ; *Neuwald, 2017*). A copy is archived at https://github.com/elifesciences-publications/bpps-sipris-code.. The fatax program annotates sequences by phylum and kingdom based on the National Center for Biotechnology Information (NCBI) taxdump and prot.accession2taxid files, available at ftp:// ftp.ncbi.nlm.nih.gov/. MAPGAPS searches were performed on the NCBI nr, env_nr and translated est databases (April 8, 2016 releases). Modeled hydrogen atoms were added to structural coordinate files using the Reduce program (*Word et al., 1999*) (http://kinemage.biochem.duke.edu/software/reduce.php).

### MAPGAPS search and alignment

MAPGAPS (*Neuwald, 2009*) creates an MSA by: (1) Taking as input either a small but ideally very accurate MSA, each sequence of which represents a distinct subgroup within a protein superfamily, or, alternatively, a set of hierarchically aligned MSAs, each of which represents a distinct subgroup. For the analyses here, we obtained from the NCBI conserved domain database (CDD) a set of hierarchically aligned MSAs or, if unavailable, a single curated MSA. A hiMSA from a previous BPPS analysis may also be used. (2) Creating a hidden Markov model (HMM) profile for each subgroup based on the input MSA. (3) Searching a protein sequence database and aligning each significantly scoring sequence (i.e. with $p \leq 0.001$) to the profile yielding the highest score. (4) Multiply aligning all the sequences obtained in this way using an alignment among profiles as a template (*Neuwald, 2009*). This process creates a large MSA that generally preserves the accuracy of the input alignment; BPPS uses this MSA as input. *Table 2* describes the structural diversity of proteins with known structure identified in this way and included in our analysis. For a superfamily of domains near the limit of current sequence analysis methods' ability to identify as related, we find that an average RMSD of 3.75 Å is typical. The RMSDs for the GNAT, EEP and UDG/TDG superfamilies fall below this value. Those

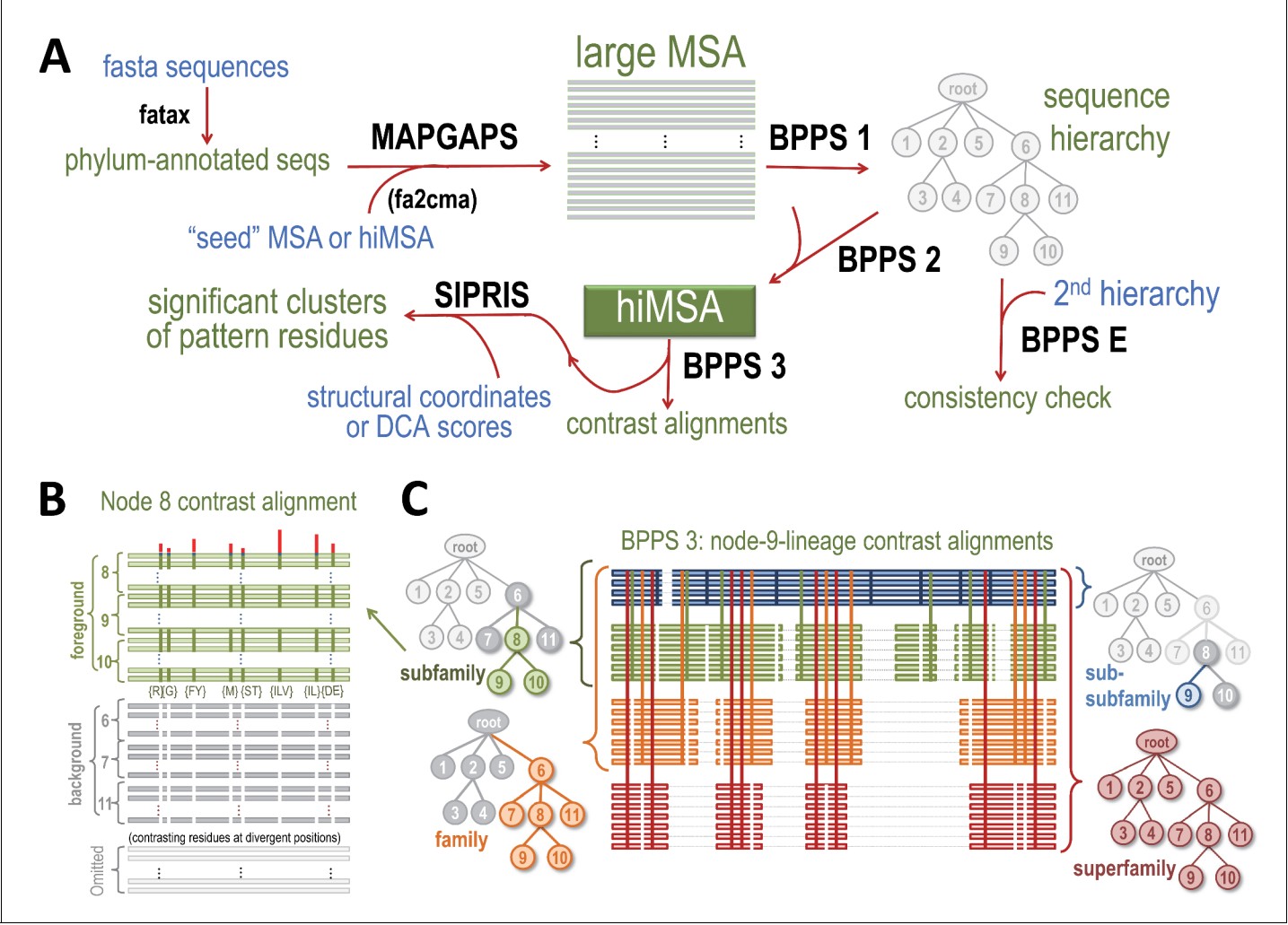

**Figure 7.** Overview of BPPS-SIPRIS analysis. (**A**) Steps required for a BPPS-SIPRIS analysis. The fatax program adds phylum-annotations to database sequences. MAPGAPS detects and aligns database sequences containing the domain defined by a cma-formatted MSA or hiMSA. (MAPGAPS can also convert an MSA from fasta- to cma-format.) This creates an MSA that step 1 of BPPS then partitions hierarchically into subgroups based on discriminating pattern residues, as illustrated schematically in (**B**). Step E of BPPS checks for consistency between BPPS step 1 runs. Step 2 of BPPS adjusts the sub-alignment for each subgroup to align and possibly assign pattern residues to regions uniquely conserved in that subgroup, thereby creating a hiMSA. Step 3 of BPPS creates, for each node in the hiMSA, lineage-specific 'contrast alignments', as is illustrated schematically in (**C**), and a corresponding input file to SIPRIS, which identifies statistically significant structural interaction networks associated with pattern residues. For further descriptions, see text. (**B**) Schematic diagram of the node eight contrast alignment. Sequences assigned to node 8's subtree (green subfamily nodes in (**C**)) constitute a 'foreground' partition; sequences assigned to the other nodes of the subtree rooted at the parent of node 8 (gray subfamily nodes in (**C**)) constitute a 'background' partition, and the remaining sequences constitute a non-participating partition. Green horizontal bars in (**B**) represent foreground sequences. The green vertical bars in (**B**) represent conserved foreground residue patterns (as shown below each bar); these diverge from (or contrast with) the background compositions at those positions (white vertical bars). Red vertical bars above quantify the degree of divergence. (**C**) Schematic diagram of a BPPS-3-generated set of 'contrast alignments' corresponding to the node 9 lineage of the sequence hierarchy in (**A**). Within a hiMSA, there is one such lineage for each leaf node. Horizontal lines represent aligned sequences and are colored by level in the hierarchy. Thin light gray horizontal lines represent non-homologous and deleted regions. Vertical lines represent the contrasting pattern positions upon which the hierarchy is based and are similarly colored by levels. The trees shown correspond to each subgroup along the lineage. The colored, gray and white nodes in each tree correspond, respectively, to their alignment foreground, background and non-participating partitions. The background for the entire superfamily (lower right) consists of standard amino acid frequencies at each position.

DOI: https://doi.org/10.7554/eLife.29880.016

**Table 2.** Structural diversity among proteins identified and aligned by MAPGAPS.

| Superfamily | structures* | | RMSD[†] (Å) | | | | Domain length[‡] | | | Resolution (Å) | |
|---|---|---|---|---|---|---|---|---|---|---|---|
| | % ID | No. | Avg | Min | Max | S.D. | MSA | Avg | S.D. | Avg | Max |
| GNAT | 27 | 16 | 3.25 | 1.0 | 6.7 | 1.4 | 125 | 139.8 | 17.0 | 1.94 | 2.61 |
| GTPases | 30 | 20 | 3.96 | 0.6 | 14.7 | 3.5 | 164 | 195.9 | 41.6 | 2.31 | 3.10 |
| Helicases | 40 | 12 | 6.39 | 2.6 | 9.8 | 1.8 | 466 | 482.8 | 60.7 | 2.86 | 3.56 |
| EEP | 40 | 16 | 3.02 | 0.8 | 5.2 | 0.95 | 241 | 259.0 | 27.6 | 2.07 | 2.99 |
| UDG/TDG | 40 | 8 | 2.54 | 1.1 | 3.6 | 0.69 | 125 | 135.9 | 12.7 | 1.83 | 2.58 |

*NMR and poor resolution structures were not used; no two proteins in each set contained more than the indicated level of percent sequence identity (% ID); pdb identifies for these are given in **supplementary file 1**.

[†]RMSDs were computed using MUSTANG (**Konagurthu et al., 2006**) with default parameters; the structural coordinates used for the analysis were limited to the domain of interest.

[‡]The number of aligned columns in the MSA, and the average length and standard deviation of the domain 'footprint'.

DOI: https://doi.org/10.7554/eLife.29880.017

for the GTPases are slightly higher, which can easily be explained by the conformational variability arising from GTPases' function as switches. The helicases yield unusually high RMSDs, which are likely due to the large domain-domain movements typical of this clade.

## MAPGAPS query alignments

Curated MSAs used for MAPGAPS searches were constructed as follows: The GNAT and GTPase MSAs were curated by L. Aravind's and A. Neuwald's group, respectively. The NCBI CDD resource group curated the other query MSAs; the CDD codes are: cd00046, DEAD-like helicase superfamily; cd08372, Exonuclease-Endonuclease-Phosphatase (EEP) domain superfamily; and cd09593, Uracil-DNA glycosylases (UDG) and related enzymes. Using these MSAs as MAPGAPS queries, we searched the NCBI nr, env_nr and translated EST databases for matching sequences. For ESTs, we obtained organism codon usage and taxonomic information from NCBI taxdump files.

## BPPS sampling

Step 1 of the BPPS (**Neuwald, 2014a**, **Neuwald, 2014b**) program stochastically partitions an MSA into hierarchically arranged subgroups (i.e. nodes). Starting from a single root node, it attaches or removes leaf nodes, moves subtrees, inserts or deletes internal nodes, moves sequences between nodes, and modifies the 'discriminating' pattern for each node. BPPS samples from among possible patterns for each subgroup based on how well each pattern distinguishes subgroup-assigned sequences (termed 'foreground' sequences) from sequences assigned to the rest of the parent node's subtree (termed 'background' sequences); *Figure 7B* illustrates this schematically. An optional Step E checks for consistency between BPPS Step 1 runs. Step 2 of BPPS (**Neuwald and Altschul, 2016a**) uses a combination of multiple sequence alignment and BPPS MCMC sampling. The G̲ibbs S̲ampler for M̲ulti-alignment O̲ptimization (GISMO) (**Neuwald and Altschul, 2016b**) adjusts each sub-group's alignment by adding regions conserved in the subgroup but not in the superfamily as a whole. Further BPPS sampling then adjusts subgroup sequence and pattern assignments taking into consideration these newly aligned regions. This converts the MSA into a hierarchical interrelated MSA (hiMSA) (*Figure 7C*). Step 3 creates, for individual nodes in the hiMSA, both a rich text formatted (rtf) contrast alignment (as shown, for example, in figure source data files) and corresponding SIPRIS input files. *Table 3* summarizes results for the five superfamilies analyzed here.

## SIPRIS

SIPRIS relies on a statistical approach termed Initial Cluster Analysis (ICA), which addresses the following questions: Consider a string of 0 s and 1 s of length $L$ and containing $D$ 1 s. Are some or all of the 1 s significantly clustered near the start of the sequence, and, if so, how surprising is the most significant such clustering? Elsewhere we describe and validate ICA (**Altschul and Neuwald, 2017**), which has a variety of biomedical applications. Here, we focus on the statistical and information theoretical bases of ICA as applied to BPPS-SIPRIS analysis.

**Table 3.** Summary of BPPS results for five superfamilies.

| Superfamily | Subgroup | # Sequences | % Identity[*] | # Nodes in subtree | Minimum subtree size |
|---|---|---|---|---|---|
| GNAT | | 237,359 | 98 | 44 | 200 |
| | Gna1 family | 1243 | | 1 | |
| GTPases | | 127,418 | 95 | 121 | 500 |
| | $R^4$ family | 18,901 | | 26 | |
| | Rab subfamily | 7002 | | 12 | |
| | Rab8 sub-subfamily | 3.312 | | 7 | |
| | TF family | 25,224 | | 10 | |
| | EFTu/CysN subfamily | 4429 | | 3 | |
| Helicases | | 131,321 | 98 | 47 | 300 |
| | RNA helicases | 36,788 | | 8 | |
| EEP | | 45,799 | 99 | 166 | 100 |
| | exoIII-AP-endo | 13,711 | | 47 | |
| | INPP5 | 3855 | | 14 | |
| TDG/UDG | | 23,592 | 98 | 47 | 100 |
| | TDG | 1639 | | 6 | |
| | UDG | 376 | | 1 | |

[*]The maximum % identity allowed between any two sequences in the set

DOI: https://doi.org/10.7554/eLife.29880.018

## BPPS-defined residue sets

Step 2 of BPPS generates a hiMSA (*Figure 7*). For each subgroup (i.e. subtree) *G* within a hierarchy, BPPS defines a corresponding set of 'discriminating' residues that most distinguish members of that subgroup from closely related subgroups. This set is ordered from the most to the least distinguishing residues. We assume that these residues are likely responsible for functions specific to subgroup *G*. Although such a set typically includes residues with well-characterized functions, our focus is on residues of unknown functional relevance. When mapped to available structures, these distinguishing residues may readily suggest plausible hypotheses; in this respect, a BPPS analysis is informative by itself. However, SIPRIS can obtain deeper insight into and corroboration of a BPPS analysis by identifying significant overlap between BPPS-defined discriminating residues and structurally defined residue sets; we term the intersection of two such sets a BPPS-SIPRIS cluster. SIPRIS analysis was motivated, in part, by Karlin and Zhu's approach (*Karlin and Zhu, 1996*) for identifying significant clusters of residues that share physical-chemical properties.

## BPPS-SIPRIS predefined clusters

The simplest BPPS-SIPRIS analysis is based on a specific, predefined structural cluster of *n* residues. This corresponds to a ball-in-urn problem, in which the BPPS-defined distinguishing residues correspond to $N_1$ red balls, the remaining residues to $N_2$ black balls, and the cluster to *n* balls drawn from the urn. The probability that at least *x* of the *n* residues are distinguishing (i.e. are 'red') is given by the cumulative hypergeometric distribution:

$$P(x,n,N_1,N_2) = \left[ \sum_{i=max(x,n-N_2)}^{min(n,N_1)} \binom{N_1}{i} \binom{N_2}{n-i} \right] \div \binom{N_1+N_2}{n}$$

## BPPS-SIPRIS optimized-clusters

Similar to BPPS-predefined clustering is choosing the optimal BPPS-structural cluster among various alternatives. To construct these, we start from a well-defined position in space, and sequentially add 'structurally adjacent' residues (variously defined, as described in Results) to generate a set of nested, structurally defined clusters. From this nested set, we select the structural cluster that optimally overlaps with the BPPS-defined residue set by applying the Minimum Description Length

(MDL) principle (*Grunwald, 2007*), as described in the next section. Optimizing over different starting residues, or different numbers of discriminating residues, requires further p-value adjustment, for which we currently apply the overly-conservative Bonferroni correction to obtain an upper bound.

## The MDL principle

To avoid overfitting BPPS-SIPRIS statistical models to observed data, we apply the MDL principle (*Grunwald, 2007*), which can be understood as formalizing Occam's Razor ('a model should not be needlessly complex'). Conceptually, this principle claims that the best among a set of alternative models is that which minimizes the description length of the model, plus the maximum-likelihood description length of the data given the model. This approach accounts for the implicit number of independent tests performed when optimizing the parameters of a model, and strikes a balance between a model's complexity and its ability to fit the data—in our case to describe biologically relevant amino acid residue patterns. More formally, a *theory* is a probability distribution over all possible sets of data, and a *model* is a parameterized set of theories. The description length of the data $D$ given a model $M$, is then defined by $DL(D|M) \equiv -\log P(D|T)$, where $T$ is maximum-likelihood theory contained in $M$ (i.e. the theory which yields the greatest probability for $D$). The description length of the model $M$ is defined by $DL(M) \equiv \log(N)$, where $N$ is the number of the effectively distinct theories (i.e. parameter settings) $M$ accommodates (*Grunwald, 2007*). The MDL principle aims to minimize $DL(D|M)+DL(M)$.

## MDL applied to BPPS-SIPRIS clustering

BPPS-optimized clustering presents several mathematical challenges. Computing valid p-values requires adjusting for the multiple tests implicit in optimizing over starting residues and clusters. Also, this optimization itself may carry an implicit bias favoring small or large clusters, as outlined below.

We start with a null model in which discriminating residues (e.g. defined by BPPS) are distributed randomly throughout an entire sequence. Given a fixed number of discriminating residues, this model yields a uniform likelihood for all sets of data, and serves as a basis of comparison for likelihoods generated by an alternative model. This model divides the sequence into an initial segment of length $x$ (which we refer to as a cluster) having $m$ discriminating residues, and a terminal segment of length $y$ having $n$ discriminating residues. The model assumes discriminating residues are generated with different probabilities in the initial and terminal segments, and its maximum-likelihood theory assigns the likelihood $p = (m/x)^m((x-m)/x)^{x-m}(n/y)^n((y-n)/y)^{y-n}$ to the data. For a particular cut-point $x$, this likelihood requires choosing the discriminating-residue probabilities $m/x$ and $n/y$ for the initial and terminal segments, and is easily normalized for the selection of these parameters. Our aim, however, is to pick the $x$ (i.e. cluster) that yields the greatest likelihood for the data. Applying the MDL principle requires calculating the effective number of independent tests $N$ implicit in choosing $x$ (*Altschul and Neuwald, 2017*). By treating $x$ as a continuous as opposed to a discrete parameter, we are able to calculate its Fisher information (*Altschul and Neuwald, 2017*), and thus $N$.

One subtlety is that simply choosing the cut point $x$ yielding the greatest likelihood implicitly favors low or high values of $x$. This occurs because the Fisher information is greater at extreme values of $x$, implying that the likelihoods are more independent of one another at those values. Empirical analyses show that this bias toward large and small clusters often yields suboptimal results from a biological perspective. However, by adding an $x$-dependent correction, derived from the Fisher information, to our optimization, we may flatten the implicit prior associated with $x$ (*Altschul and Neuwald, 2017*). Random simulation shows that analytic p-values computed using our approach fall within about 20% of empirical p-values. We still need to adjust these p-values for clusters found using different starting residues. Absent a better approach, we currently apply the simple but overly conservative Bonferroni correction (*Bonferroni, 1936*).

## Runtimes

The runtime bottleneck in an analysis is BPPS. BPPS runtimes depend on the desired depth of the hierarchy, on the width of and the number of sequences in the input MSA and on the minimum number of sequences required to define a leaf node. For example, on a 64-bit Linux workstation, a 125,000-sequence GTPase MSA requires about 4 weeks to generate a 120 node hierarchy up to

eight nodes deep and with a minimum leaf node size of 500 sequences. Note that much of this time is spent marginally refining a hierarchy. This approach is not recommended. Instead, we suggest running an initial analysis at a depth of 1 and then using the BPPS 'focus' option with a maximum depth of 2–4 to expand the subtree for a specific major node of interest. For the GTPase MSA, this approach takes less than a few days.

## MSA cma format

The programs used here require cma-formatted MSAs. The cma (collinear multiple alignment) format, which is unique to our programs, allows the specification of a hierarchically-arranged set of MSAs, such as are created in step 2 of BPPS and which serve as input to the MAPGAPS program. (MAPGAPS will also take as input a single MSA either in cma or fasta format.) For a single MSA, the cma format consists of a header line, such as '[0_(1)=name(135){go = 10000,gx = 2000,pn = 1000.0, lf = 0,rf = 0}:'. The leftmost '0' labels this as the root node of a hierarchical MSA; '(1)' indicates a single aligned block (this parameter is utilized during MCMC sampling); 'name' labels the MSA; '135' indicates the number of aligned sequences; and the string in curly brackets gives parameter settings that are not used here. This is followed by a second header line, such as '(20)********************', where '20' indicates the number of aligned columns and the asterisks designate which columns MCMC column should be sampled (*Neuwald et al., 1997*).

Each sequence in the MSA is specified by three lines. An example of the first line is '$41 = 34 (28):', where '$41' indicates that this is the 41$^{st}$ sequence, '34' indicates the total number of residues in the sequence and '28' the number of residues and gaps ('-') minus the number of insertions (this information is required for MCMC sampling). The second line gives a fasta formatted identifier and description, such as '>4ABC_A'. And the third line, such as '{(QEYP)ID-QTGKCEPYigqiTKCStfLPNST (NVTN)}*', specifies the aligned sequence where residues within parentheses represent regions flanking the aligned region on either side; upper- and lower-case letters represent matching and insertion residues, respectively; and gap characters represent deletions. The curly brackets on each end allow multiple aligned blocks to be defined during MCMC sampling. The last line of the MSA, such as '_0].', indicates the end of the MSA; this syntax allows multiple (hierarchically arranged) MSAs to be included within a single input file.

## Additional considerations

BPPS assigns a log-odds score to each pattern residue; ranked by these scores, a specific number of positions are considered by SIPRIS. SIPRIS identifies the statistically most significant intersection, if any, between the BPPS- and structurally defined residue sets; adjusting its p-value for the number of starting residues considered. Note that discriminating residues outside of the intersection may have BPPS scores as high as or higher than those within; SIPRIS makes no distinctions in this regard.

## PyMol 3D visualization

Given structural coordinates as input, the BPPS and SIPRIS programs will generate PyMol (*Schrodinger, 2010*) scripts to aid visualization of BPPS-defined residues and of BPPS-SIPRIS structural networks, respectively.

## Acknowledgements

LA and SFA were supported by the Intramural Research Program of the National Institutes of Health, National Library of Medicine. AFN received no specific funding for this work, but was supported by the University of Maryland, Baltimore. The funders had no role in study design, data collection and analysis, decision to publish, or preparation of the manuscript.

## Additional information

### Funding

| Funder | Grant reference number | Author |
| --- | --- | --- |
| University of Maryland | | Andrew F Neuwald |

| National Institutes of Health | Intramural Research Program | L Aravind Stephen F Altschul |

The funders had no role in study design, data collection and interpretation, or the decision to submit the work for publication.

### Author contributions
Andrew F Neuwald, Conceptualization, Data curation, Software, Formal analysis, Validation, Investigation, Visualization, Methodology, Writing—original draft, Project administration, Writing—review and editing; L Aravind, Formal analysis, Validation, Investigation, Writing—original draft, Writing—review and editing; Stephen F Altschul, Conceptualization, Formal analysis, Methodology, Writing—original draft, Writing—review and editing

### Author ORCIDs
Andrew F Neuwald (iD) http://orcid.org/0000-0002-0086-5755

### Decision letter and Author response
Decision letter https://doi.org/10.7554/eLife.29880.029
Author response https://doi.org/10.7554/eLife.29880.030

# Additional files

## Supplementary files
• Supplementary file 1. The pdb files used for computing RMSDs in *Table 2*.
DOI: https://doi.org/10.7554/eLife.29880.019

• Transparent reporting form
DOI: https://doi.org/10.7554/eLife.29880.020

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

## Appendix 1

DOI: https://doi.org/10.7554/eLife.29880.021

# BPPS-SIPRIS and functional binding sites

BPPS-SIPRIS does not seek to identify functional binding sites per se, though it often reveals such sites after the fact. We illustrate this by comparing our analyses of Gna1, Rab4 and Rab8 with annotations on the Inferred Biomolecular Interactions Server (IBIS) (*Shoemaker et al., 2012*)—keeping in mind that BPPS-SIPRIS will also identify residues having other roles. Thirteen of the 22 residues in the Gna1 family-specific network are among the 27 residues annotated on the IBIS as binding either to substrate (N-acetyl-D-glucosamine-6-phosphate) or to the other homodimeric subunit. Eleven of the 14 residues in the Rab4 subfamily-specific network are among the 15 Rabenosyn-5 binding residues annotated by the IBIS. Structures are available for Rab8 bound to Ocrl1 (3qbt) (*Hou et al., 2011*), to the guanine nucleotide exchange factor Rabin8 (4lhx) (*Guo et al., 2013*), and to the bMERB domain of Mical-cL (5szi) (*Rai et al., 2016*). The IBIS annotates nine Rab8 residues as binding to all three components: K10, I43, D44, F45, I47, W62, I73, Y77, and R79. Seven of these (all but the flanking residues, K10 and R79) are included among the 19 Rab8 subfamily and sub-subfamily residues that interact with the Ocrl1 binding interface defined by SIPRIS. Of course, SIPRIS performs a benchmarking role similar to that of the IBIS inasmuch as both utilize 3D interactions within high quality crystal structures as gold standards.

# Benchmarking BPPS against the SFLD

We benchmarked BPPS by applying it to eight superfamilies within the Structure Function Linkage Database (SFLD) (*Akiva et al., 2014*); other SFLD superfamilies contained too few sequences or subgroups (<2) for BPPS. SFLD classifies each superfamily into subgroups based on sequence and structural considerations and, to a very limited extent, on experimental data. Most SFLD superfamilies also include unannotated and automatically annotated sequences, which were used to enlarge the input sets, but not for benchmarking. We aligned the sequences for each superfamily using GISMO (*Neuwald and Altschul, 2016b*), and then removed both redundant (>98% identical) sequences and sequences having deletions in more than 30% of aligned columns. To ensure sufficient statistical support, we required that each BPPS leaf node correspond to at least 100 to 800 sequences, depending on superfamily size and diversity.

BPPS misclassified at most 0.14% of the annotated sequences overall (*Appendix 1—table 1*). For some superfamilies, BPPS assigned some subgroups to multiple nodes or several subgroups to a single node; in such cases, however, there were no inherent conflicts. As illustrated in *Appendix 1—table 2*, for haloacid dehalogenases (HADs): (i) BPPS assigned certain SFLD subgroups (SGs) to the root node, presumably due to their being too small or lacking a significant distinguishing pattern. (ii) For certain SFLD subgroups BPPS assigns some sequences to the root node due to a significant number of pattern mismatches (e.g., SG1138). (iii) BPPS misclassified 101 HAD sequences, probably due to misalignments. (This MSA contains a large number of indels rendering it relatively inaccurate—a problem that is addressed in Materials and Methods.) (iv) Several sequences appear to be erroneously assigned by SFLD to SG1129 (*Appendix 1—table 3*). BPPS assigned 108 of the 129 sequences in SG1129 to the root, but 21 sequences to four other nodes. As shown in *Appendix 1—table 3*, using the criterion of mean BLAST score, the sequences assigned to each node better match non-SG1129 sequences of that node than they do SG1129 sequences assigned to other nodes. This analysis avoids problems that may arise from MSA errors and suggests how best to reassign misclassified sequences (*Appendix 1—figure 1* and *Appendix 1—table 4*). A similar analysis of the radical SAM superfamily indicates that 326 sequences assigned by SFLD to SG1118 are more closely related to SG1066 than they are to other SG1118 sequences. The respective high mean BLAST scores of 429 and 210 may suggest that SG1118 and SG1066 should be merged. In this way, BPPS may be useful for protein subgroup database curation.

**Appendix 1—table 1.** Summary of SFLD benchmarking of BPPS.

| Superfamily | # subgroups | | BPPS min.* | Annotated by SFLD | | | BSG‡ | BPPS conflicts§ | | | Maximum % errors# |
|---|---|---|---|---|---|---|---|---|---|---|---|
| | SFLD | BPPS | | No | Yes | expt† | | Error | ? | Correct | |
| radical SAM | 49 | 17 | 800 | 52,608 | 17,680 | 12 | 13,676 | 10 | 6 | 326 | 0.12 |
| glutathione transferase | 26 | 15 | 100 | 6921 | 3633 | 0 | 1945 | 0 | 0 | 0 | 0 |
| peroxiredoxin | 6 | 11 | 100 | 3870 | 5521 | 0 | 5255 | 0 | 1 | 0 | 0.02 |
| haloacid dehalogenase | 24 | 28 | 200 | 21,768 | 33,379 | 9 | 26,589 | 35 | 66 | 27 | 0.38 |
| isoprenoid synthase I | 9 | 7 | 200 | 9666 | 1604 | 55 | 1536 | 0 | 0 | 0 | 0 |
| isoprenoid synthase II | 3 | 5 | 100 | 6974 | 671 | 38 | 591 | 1 | 0 | 0 | 0.17 |
| nitroreductase | 110 | 11 | 200 | 0 | 17,318 | 0 | 7242 | 20 | 11 | 0 | 0.43 |
| enolase | 8 | 8 | 800 | 26,227 | 2267 | 7 | 2143 | 0 | 0 | 0 | 0 |
| | | | total: | | 82,073 | 121 | 58,977 | 66 | 84 | 353 | avg: 0.14 |

*The minimum number of sequences required for each BPPS subgroup.

†Numbers of experimentally validated annotations.

‡The number of SFLD annotated sequences assigned by BPPS to a subgroup.

§The number of SFLD annotated sequences in conflict with BPPS classification; error, SFLD annotation appears to be correct; correct, BPPS appears to be correct

#Percent erroneous or ambiguous ('?') BPPS assignments among annotated sequences not assigned to a root node.

DOI: https://doi.org/10.7554/eLife.29880.023

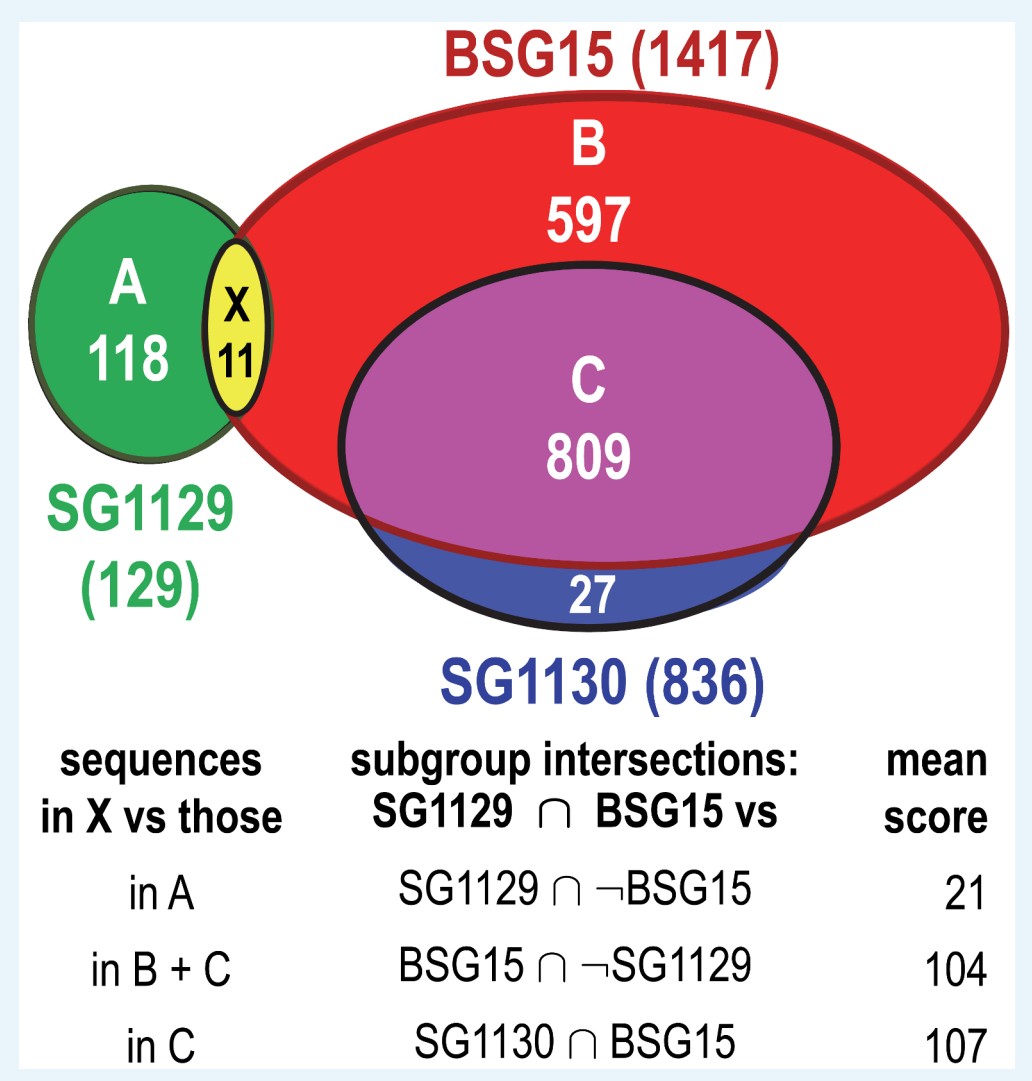

| sequences in X vs those | subgroup intersections: SG1129 ∩ BSG15 vs | mean score |
|---|---|---|
| in A | SG1129 ∩ ¬BSG15 | 21 |
| in B + C | BSG15 ∩ ¬SG1129 | 104 |
| in C | SG1130 ∩ BSG15 | 107 |

**Appendix 1—figure 1.** Eleven haloacid dehalogenase sequences that the SFLD assigned to SG1129, but that are more closely related to SG1130 sequences. The Venn diagram shows the overlap between the subgroups BSG15, SG1129 and SG1130 with the numbers of sequences indicated. The table gives the mean pairwise gapped BLAST scores for the 11 sequences assigned to both SG1129 and BSG15 versus the sequence sets shown; this analysis indicates that the 11 sequences should be reassigned from SG1129 to SG1130. Similar analyses indicate that four other sequences in SG1129 should be reassigned to SG1135 (based on mean scores of 27 versus 139) and that a sequence in SG1136 should be reassigned to SG1137 (based on a mean score of 8 versus 149).

DOI: https://doi.org/10.7554/eLife.29880.022

**Appendix 1—table 2.** Correspondence between BPPS and SFLD subgroups for haloacid dehalogenases*.

| Subgroup IDs | | SFLD& | SFLD[#] | |
|---|---|---|---|---|
| BPPS | SFLD[‡] | BPPS[§] | Total | % |
| root[†] | various | 1531 | 1618 | 96.2 |
| | 1138 | 82 | 833 | 9.8 |

*Appendix 1—table 2 continued on next page*

Appendix 1—table 2 continued

| Subgroup IDs | | SFLD& | SFLD# | |
|---|---|---|---|---|
| BPPS | SFLD‡ | BPPS§ | Total | % |
| 34 | 0 | 200 | 21768 | 0.9 |
| | **1129** | **3** | **129** | **2.3** |
| 23 | 0 | 101 | 21768 | 0.5 |
| | *1124* | *1* | *495* | *0.2* |
| | 1135 | 125 | 9423 | 1.3 |
| 21 | 0 | 158 | 21768 | 0.7 |
| | 1124 | 91 | 495 | 18.4 |
| | *1145* | *2* | *43* | *4.7* |
| 20 | 0 | 46 | 21768 | 0.2 |
| | 1131 | 162 | 201 | 80.6 |
| 25 | 0 | 76 | 21768 | 0.3 |
| | 1135 | 311 | 9423 | 3.3 |
| 2 | 0 | 1915 | 21768 | 8.8 |
| | 2 | 10091 | 11846 | 85.2 |
| 3 | 0 | 937 | 21768 | 4.3 |
| | **1129** | **4** | **129** | **3.1** |
| | *1134* | *1* | *866* | *0.1* |
| | 1135 | 4500 | 9423 | 47.8 |
| | *1139* | *4* | *1851* | *0.2* |
| | *1140* | *1* | *821* | *0.1* |
| 4 | 0 | 2422 | 21768 | 11.1 |
| | *2* | *1* | *11846* | *0.0* |
| | *1137* | *9* | *1430* | *0.6* |
| | <u>1140</u> | <u>3</u> | <u>821</u> | <u>0.4</u> |
| | 1141 | 53 | 278 | 19.1 |
| | *1142* | *2* | *236* | *0.8* |
| | 1144 | 2497 | 2759 | 90.5 |
| 5 | 0 | 229 | 21768 | 1.1 |
| | 2 | 986 | 11846 | 8.3 |
| 6 | 0 | 342 | 21768 | 1.6 |
| | 1124 | 360 | 495 | 72.7 |
| 7 | 0 | 330 | 21768 | 1.5 |
| | 1134 | 628 | 866 | 72.5 |
| 33 | 0 | 100 | 21768 | 0.5 |
| | 1134 | 153 | 866 | 17.7 |
| 8 | 0 | 57 | 21768 | 0.3 |
| | 1133 | 400 | 400 | 100 |
| 32 | 0 | 32 | 21768 | 0.1 |
| | 1139 | 218 | 1851 | 11.8 |
| 31 | 0 | 28 | 21768 | 0.1 |
| | 1139 | 216 | 1851 | 11.7 |
| 9 | 0 | 195 | 21768 | 0.9 |

*Appendix 1—table 2 continued*

| Subgroup IDs | | SFLD& | SFLD# | |
|---|---|---|---|---|
| BPPS | SFLD‡ | BPPS§ | Total | % |
| | *1134* | *1* | *866* | *0.1* |
| | 1139 | 942 | 1851 | 50.9 |
| 10 | 0 | 105 | 21768 | 0.5 |
| | 1137 | 896 | 1430 | 62.7 |
| 11 | 0 | 284 | 21768 | 1.3 |
| | **1135** | **3** | **9423** | **0.0** |
| 12 | 0 | 478 | 21768 | 2.2 |
| | **1136** | **1** | **246** | **0.4** |
| | 1137 | 178 | 1430 | 12.4 |
| 13 | 0 | 117 | 21768 | 0.5 |
| | 1138 | 751 | 833 | 90.2 |
| 14 | 0 | 1034 | 21768 | 4.8 |
| | 1135 | 32 | 9423 | 0.3 |
| 15 | 0 | 525 | 21768 | 2.4 |
| | **1129** | **11** | **129** | **8.5** |
| | 1130 | 809 | 836 | 96.8 |
| | *1132* | *6* | *227* | *2.6* |
| | <u>1135</u> | <u>63</u> | <u>9423</u> | <u>0.7</u> |
| | *1139* | *3* | *1851* | *0.2* |
| 16 | 0 | 337 | 21768 | 1.5 |
| | *1135* | *1* | *9423* | *0.0* |
| | 1140 | 670 | 821 | 81.6 |
| 17 | 0 | 230 | 21768 | 1.1 |
| | 1135 | 288 | 9423 | 3.1 |
| 18 | 0 | 505 | 21768 | 2.3 |
| | 1135 | 950 | 9423 | 10.1 |
| 19 | 0 | 197 | 21768 | 0.9 |
| | *1129* | *3* | *129* | *2.3* |
| 22 | 0 | 338 | 21768 | 1.6 |
| 24 | 0 | 110 | 21768 | 0.5 |
| | 1135 | 107 | 9423 | 1.1 |

*Erroneous, ambiguous and corrected classifications are shown as italicized, underlined, and bold, respectively.

†Averages over 12 root-assigned subgroups.

‡SFLD subgroups represented in each BPPS subgroup; zero indicates the SFLD unannotated sequence set.

§The number of sequences in both the SFLD and BPPS subgroups in each row.

#Total number of sequences in each SFLD subgroup and the percentage of these in the BPPS subgroup.

DOI: https://doi.org/10.7554/eLife.29880.024

## Dependency of BPPS-SIPRIS on the input MSA

We assessed the dependence of BPPS-SIPRIS on the quality of the input using jackhmmer (*Finn et al., 2015*) MSAs, which should be less accurate than the MAPGAPS MSAs used here (see Materials and methods and *Figure 7*). Jackhmmer, which applies a query-centric iterative algorithm similar to that of PSI-BLAST (*Altschul et al., 1997*), was run over the EV-

**Appendix 1—table 3.** Haloacid dehalogenase SG1129 sequences that BPPS assigned to distinct subgroups (BSG).

| BPPS BSG | SG1129 # seqs | In BSG* | % matches to BPPS pattern† BSG 34 | BSG 3 | BSG 15 | BSG 19 | Mean score vs other seqs:‡ In SG1129 | In BSG |
|---|---|---|---|---|---|---|---|---|
| 34 | 203 | 3 | 80 | 31 | 37 | 8 | 14 | 399 |
| 3§ | 5447 | 4 | 6 | 96 | 55 | 5 | 27 | 137 |
| 15# | 1417 | 11 | 10 | 50 | 87 | 12 | 21 | 104 |
| 19 | 200 | 3 | 8 | 21 | 36 | 83 | 9 | 488 |
| root | 16,869 | 108 | 9 | 40 | 42 | 8 | na | na |

*The number of SG1129 sequences assigned to the BSG in each row.

†Average percentage of matches to the pattern residues for their assigned BSG among the SG1129 sequences. The highest percentages (bold) correspond to each BSG's own pattern.

‡The mean pairwise BLAST scores of the reassigned sequences against the remaining sequences either in SG1129 or in the BSG for that row.

§This BSG corresponds to SG1135. (See **Appendix 1—table 2.**)

#This BSG corresponds to SG1130. (See **Appendix 1—table 2.**)

DOI: https://doi.org/10.7554/eLife.29880.025

**Appendix 1—table 4.** Average percentage of matches to various BPPS subgroup (BSG) patterns for haloacid dehalogenase sequences assigned to SFLD subgroup SG1135.

| BSG | | SG1135 | % matches to each BSG pattern for SG1135 sequences*: | | | | | | | | | | Mean score[†] vs others | | new[‡] |
|---|---|---|---|---|---|---|---|---|---|---|---|---|---|---|---|
| ID | # seqs | # seqs | 15 | 16 | 25 | 3 | 11 | 14 | 17 | 18 | 24 | 23 | In SG1135 | In BSG | SGs |
| 15 | 1417 | 63 | **69** | 35 | 32 | 51 | 8 | 42 | 19 | 56 | 22 | 20 | 63 | **64** | ? |
| 16 | 1008 | 1 | 43 | 40 | 52 | 52 | 12 | 40 | 40 | 52 | 24 | 8 | **110** | 14 | error |
| 25 | 387 | 311 | 48 | 37 | **84** | 57 | 13 | 36 | 43 | 45 | 27 | 10 | 80 | **309** | yes |
| 3 | 5447 | 4500 | 49 | 25 | 35 | **90** | 15 | 37 | 20 | 38 | 20 | 9 | **148** | 132 | ? |
| 11 | 287 | 3 | 33 | 29 | 37 | 51 | **93** | 32 | 17 | 32 | 16 | 8 | 44 | **632** | yes |
| 14 | 1066 | 32 | 49 | 27 | 25 | 37 | 9 | **77** | 18 | 47 | 19 | 13 | 42 | **130** | yes |
| 17 | 518 | 288 | 41 | 39 | 41 | 42 | 8 | 30 | **88** | 48 | 25 | 6 | 55 | **321** | yes |
| 18 | 1455 | 950 | 58 | 34 | 28 | 44 | 9 | 50 | 16 | **88** | 20 | 15 | 43 | **169** | yes |
| 24 | 217 | 107 | 42 | 35 | 26 | 44 | 9 | 32 | 20 | 42 | **92** | 4 | 53 | **293** | yes |
| 23 | 227 | 125 | 62 | 32 | 23 | 41 | 4 | 32 | 10 | 36 | 14 | **90** | 45 | **403** | yes |
| root | n.a. | 3040 | 46 | 37 | 41 | 56 | 12 | 40 | 29 | 46 | 23 | 10 | n.a. | n.a. | n.a. |

*Average percentage of matches to the pattern residues for their assigned BSG among the SG1135 sequences. The highest percentages (bold) correspond to the highest percentage in each row.

[†]The mean pairwise BLAST scores of the BPPS-assigned sequences against the remaining sequences either in SG1135 or in the BSG for that row. The highest scores in each row are bold. (See **Appendix 1—table 2.**)

[‡]A 'yes' in this column indicates that the SG1135 sequences assigned to the BSG in that row likely correspond to a subgroup distinct from SG1135; '?' indicates a possible subcategory of SG1135; 'error' indicates a BPPS misclassification.

DOI: https://doi.org/10.7554/eLife.29880.026

**Appendix 1—table 5.** BPPS-SIPRIS analyses using MAPGAPS (MG) versus Jackhmmer (JH) generated MSAs as input.

| Protein | MSA* | SIPRIS† Mode | BPPS-SIPRIS† Dist. | Init. | Term. | SIPRIS p-value | Tree level‡ | Optimal BPPS pattern ∩ SIPRIS cluster# |
|---|---|---|---|---|---|---|---|---|
| Gna1 | JH | p=BDF | 21 | 69 | 87 | $1.2 \times 10^{-5}$ | 1 | F93,I94,D105,K136,H95,Y68,Y135,T44,R102,E104,E90,C141,K92,L40,L43,**V88**,V134,Y36,**L27**,F58,V89 |
|  | MG | p=BDF | 22 | 57 | 71 | $8.5 \times 10^{-7}$ | 1 | F93,I94,D105,K136,H95,Y68,Y135,T44,R102,E104,E90,C141,K92,**M61**,L40,L43,V134,**F54**,Y36,F58,**G98**,V89 |
|  | JH | S | 14 | 21 | 135 | $2.1 \times 10^{-5}$ | 1 | E90,K92,R102,V89,**V88**,G101,Y135,I94,V134,K136,F93,E104,Y68,H95 |
|  | MG | S | 14 | 21 | 107 | $2.5 \times 10^{-4}$ | 1 | E90,K92,R102,V89,G101,Y135,I94,V134,K136,F93,**G98**,E104,Y68,H95 |
| APE1 | JH | H | 15 | 38 | 219 | $5.0 \times 10^{-5}$ | 0 | V206,L167,Q95,S66,G209,W67,**P311**,H309,D283,S307,N68,D210,E96,N212,**R185** |
|  | MG | H | 16 | 33 | 218 | $4.2 \times 10^{-7}$ | 0 | V206,L167,**F165**,Q95,S66,G209,**V69**,W67,H309,D283,**T265**,S307,N68,D210,E96,N212 |
|  | JH | H | 25 | 158 | 99 | $8.8 \times 10^{-6}$ | 1 | Y128,E154,R156,Y171,P173,W188,D70,W267,N277,E236,C310,**G71**,R237,D219,R254,**R281**,V213,A214,**L62**,G279,K98,**V131**,A175,**L72**,**R181** |
|  | MG | H | 25 | 137 | 214 | $1.7 \times 10^{-6}$ | 1 | Y128,**G155**,E154,**D152**,R156,Y171,P173,**R185**,D70,W267,N277,W188,E236,C310,**Y269**,R237,D219,R254,V213,A214,**Y264**,G279,K98,A175,**G145** |
| Rho1 | JH | B | 20 | 63 | 110 | $3.5 \times 10^{-4}$ | 1 | S106,D28,W114,Y81,Y89,A76,Q78,L84,E79,K22,W73,**F176**,F99,F107,V101,E163,Y161,G149,C153 |
|  | MG | B | 20 | 53 | 100 | $8.3 \times 10^{-5}$ | 1 | S106,D28,W114,Y81,Y89,A76,Q78,L84,E79,K22,W73,F99,F107,V101,**V144**,**R137**,E163,Y161,G149,C153 |
|  | JH | C | 24 | 82 | 91 | $2.4 \times 10^{-6}$ | 1 | L84,Y81,Q78,A76,**E117**,W114,Y89,D28,**V24**,E79,W73,K22,F99,F107,Y161,C153,G149,S106,V101,E163,G131,**F176**,**F57** |
|  | MG | C | 22 | 55 | 98 | $7.8 \times 10^{-7}$ | 1 | L84,Y81,Q78,A76,**T52**,W114,Y89,D28,E79,W73,K22,F99,F107,Y161,C153,G149,**V144**,S106,V101,E163,G131,**R137** |

*Appendix 1—table 5 continued on next page*

Appendix 1—table 5 continued

| Protein | MSA* | SIPRIS† Mode | BPPS-SIPRIS† Dist. | Init. | Term. | SIPRIS p-value | Tree level‡ | Optimal BPPS pattern ∩ SIPRIS cluster# |
|---|---|---|---|---|---|---|---|---|
| eIF4AIII | JH | p=J | 8 | 18 | 212 | $2.7 \times 10^{-4}$ | 1 | G165,R116,D169,R166,G143,T115,G196,P164 |
| | MG | p=J | 11 | 18 | 128 | $6.4 \times 10^{-6}$ | 1 | G165,**F197**,R116,**Q200**,D169,R166,G143,T115, **G142**,G196,P164 |

*Input MSA: Jackhmmer, JH; MAPGAPS, MG.

†Explained in the footnotes to **Table 1**.

‡Codes designate BPPS category: 0, superfamily; 1, family.

§Pattern residue discrepancies between the Jackhmmer and MAPGAPS runs are shown in bold.

DOI: https://doi.org/10.7554/eLife.29880.027

fold website (evfold.org) with its default parameter settings (five iterations; aligned columns or sequences with >30% deletions ignored). Using Gna1, APE1, Rho1 and eIF4AIII as queries, jackhmmer yielded MSAs of 107,738, 36,297, 99,838 and 107,213 sequences, respectively; these correspond to four of the five superfamilies examined here. (For the UDG/TDG superfamily, jackhmmer pulled in too few sequences for a valid comparison.) In each case, BPPS-SIPRIS identified nearly the same residue sets using either type of MSA (*Appendix 1—table 5*), with the *p*-values for jackhmmer MSAs tending to be less significant. Thus, BPPS-SIPRIS yields fairly consistent results using very different MSAs, although using MAPGAPS with a curated MSA helps detect more sequences and improve sensitivity.

