## [Decision Letter]

Thank you for submitting your article "Inferring joint sequence-structural determinants of protein functional specificity" for consideration by *eLife*. Your article has been reviewed by three peer reviewers, and the evaluation has been overseen by a Reviewing Editor and Richard Aldrich as the Senior Editor. The following individual involved in review of your submission has agreed to reveal his identity: David T Jones (Reviewer #2).

The reviewers have discussed the reviews with one another and the Reviewing Editor has drafted this decision to help you prepare a revised submission.

Summary:

The authors present an approach to identify functionally specific residues that distinguish functional subgroups in a protein evolutionary superfamily. The method is based on identifying clusters of residues through sequence analysis (Bayesian Partitioning With Pattern Selection, BPPS) that are distinctive of individual subfamilies of a larger family/superfamily. This is followed by a second step identifying clusters of residues using an approach based on protein structure (Structurally Interacting Pattern Residues' Inferred Significance, SIPRIS) and calculating a statistical significance for the overlap of the structure-based cluster with the sequence-based cluster. This combination is a potentially very robust, insightful, and valuable approach, with widespread applications in uncovering the mechanisms of allostery.

Essential revisions:

The reviewers were enthusiastic about the potential impact of the method, as well as the writing and presentation of the manuscript, and agreed that benchmarking of allosteric residues is limited by the available experimental literature, justifying the use of specific examples. However, other aspects of the approach require more quantification and benchmarking. Specifically, confidence in the method would be greatly improved by the inclusion of:

1) A benchmark of the ability of the BPPS/SIPRIS strategy to identify functional binding site residues, such as specificity-determining residues that distinguish protein subgroups binding different ligands. These predictions should be easy to assess by using known substrate binding sites in e.g. the IBIS resource.

2) Benchmarking of the partitioning into sub-families given by the BPPS method. The authors could compare the subgrouping achieved by the MCMC method of BPPS with groupings identified by experimental characterisation of relatives, e.g. in the SFLD resource that provides curated hierarchical groupings for superfamily sequence relatives. See for example the benchmarking of subfamily grouping in Brown, D.P., Krishnamurthy, N. and Sjölander, K., 2007. Automated protein subfamily identification and classification. PLoS computational biology, 3(8), p.e160.

3) An analysis or discussion of the dependence of the BPPS method on the quality of the input multiple sequence alignment (MSA). Although a sequence search is used to expand the datasets, presumably it is necessary to have key representatives in the starting cluster. This suggests that some knowledge of the subgroups is necessary before starting the BPPS analysis. How dependent is the method on having informative sets of sequences for each subgroup in the starting cluster? What is the range of RMSD between relatives in the MSA used to analyse the example superfamilies? How applicable will this method be to structurally very divergent superfamilies?

4) Per journal policy, the BPPS and SIPRIS software must be made maximally available. The code must conform to the Open Source Definition (https://opensource.org/docs/osd), and should be deposited in an appropriate public repository. To ensure that software can be reproduced without restrictions and that authors are properly acknowledged for their work, authors should license their code using an open source license. Authors are encouraged to use version control services such as GitHub, GitLab, and SourceForge. *eLife* maintains a GitHub account to archive code accompanying *eLife* publications that has been deposited on GitHub or another version control service.

---

## [Author Response]

Essential revisions:The reviewers were enthusiastic about the potential impact of the method, as well as the writing and presentation of the manuscript, and agreed that benchmarking of allosteric residues is limited by the available experimental literature, justifying the use of specific examples. However, other aspects of the approach require more quantification and benchmarking. Specifically, confidence in the method would be greatly improved by the inclusion of:1) A benchmark of the ability of the BPPS/SIPRIS strategy to identify functional binding site residues, such as specificity-determining residues that distinguish protein subgroups binding different ligands. These predictions should be easy to assess by using known substrate binding sites in e.g. the IBIS resource.

We report significant overlap between IBIS-inferred biomolecular interactions and BPPS-SIPRIS results for Gna1, Rab4 and Rab8. However, we note two general concerns about using IBIS as a generic benchmark for BPPS-SIPRIS. First, instead of identifying functional binding site residues per se, BPPS-SIPRIS more generally identifies clusters of residues potentially responsible for allosteric and other, as yet poorly characterized protein properties; its results typically require interpretation by the user based on structural and biochemical considerations. The identified residues may or may not interact with other cellular components. Second, a BPPS-SIPRIS analysis is most informative when applied to the largest subgroups within a superfamily. Residues most characteristic of a small subfamily of proteins, which IBIS records as binding to a specific ligand, may fail to reach a detectable level of statistical significance. In extreme cases, replacement of one or two residues may be sufficient to change substrate specificity. Therefore, using IBIS predefined residue sets to benchmark BPPS-SIPRIS is often problematic. Note too that SIPRIS and IBIS both utilize as gold standards 3D interactions within high quality crystal structures.

2) Benchmarking of the partitioning into sub-families given by the BPPS method. The authors could compare the subgrouping achieved by the MCMC method of BPPS with groupings identified by experimental characterisation of relatives, e.g. in the SFLD resource that provides curated hierarchical groupings for superfamily sequence relatives. See for example the benchmarking of subfamily grouping in Brown, D.P., Krishnamurthy, N. and Sjölander, K., 2007. Automated protein subfamily identification and classification. PLoS computational biology, 3(8), p.e160.

We describe benchmarking of SFLD superfamilies in Appendix 1. Very few of the SFLD sequence annotations are based on direct experimentation: Only 5 of the 8 superfamilies we examined contained such sequences, and the annotations for none of the 121 experimentally studied sequences conflicted with the BPPS analysis. Nearly all the SFLD annotations were derived from sequence/structural similarity. Some SFLD subgroups were assigned by BPPS to the root node, presumably because they were either too small or lacked a significant characteristic pattern. Some sequences assigned to the root belong to subgroups that for the most part were assigned by BPPS to non-root-subgroups; less stringent priors might pull these out of the root, but could also include aberrant sequences that would cloud the analysis. Ignoring subgroups and sequences that it assigned to the root (28% of the annotations), BPPS misclassified at most 0.14% of the remaining annotated sequences. Many of these errors appear due to misalignment, as the MSAs were created automatically. However, BPPS also revealed what appear to be SFLD misclassification errors (see Appendix 1—Tables 1–4 and Appendix 1—Figure 1). Thus, rather than treating SFLD as a gold standard, we base our analysis on the principle of consistency. The manuscript also cites alternative approaches to evaluating BPPS protein domain hierarchies.

3) An analysis or discussion of the dependence of the BPPS method on the quality of the input multiple sequence alignment (MSA). Although a sequence search is used to expand the datasets, presumably it is necessary to have key representatives in the starting cluster. This suggests that some knowledge of the subgroups is necessary before starting the BPPS analysis. How dependent is the method on having informative sets of sequences for each subgroup in the starting cluster? What is the range of RMSD between relatives in the MSA used to analyse the example superfamilies? How applicable will this method be to structurally very divergent superfamilies?

We describe and discuss BPPS and BPPS-SIPRIS analyses using two alternative MSA methods, GISMO and Jackhmmer. Unlike MAPGAPS, which takes as input a curated MSA, both of these methods create an MSA *ab initio*. We used GISMO to align sequences for the SFLD analysis with reasonable results (see previous response). Jackhmmer is an iterative, query-based MSA method similar to PSI-BLAST; we applied it to one protein from each of the five superfamilies examined here. In general, such query-centric methods align non-query sequences to each other less accurately than do global alignment programs. For the UDG/TDG superfamily, Jackhmmer failed to detect a sufficient number of sequences for BPPS to construct a hierarchy comparable to that obtained using MAPGAPS with a curated MSA as input. For the four other superfamilies, Jackhmmer produced an MSA which, when used as input to BPPS-SIPRIS, identified nearly the same query-family cluster of residues as did our initial analysis—though generally with lower statistical significance (Appendix 1—Table 6). Of course, currently it is not difficult to obtain curated alignments to use as input to MAPGAPS (e.g., from the NCBI CDD or PFAM). Table 2 gives the range of RMSDs between proteins of known structure. We find that BPPS-SIPRIS works well as long as the input sequences contain statistically detectable residue patterns.

4) Per journal policy, the BPPS and SIPRIS software must be made maximally available. The code must conform to the Open Source Definition (https://opensource.org/docs/osd), and should be deposited in an appropriate public repository. To ensure that software can be reproduced without restrictions and that authors are properly acknowledged for their work, authors should license their code using an open source license. Authors are encouraged to use version control services such as GitHub, GitLab, and SourceForge. eLife maintains a GitHub account to archive code accompanying eLife publications that has been deposited on GitHub or another version control service.

We have made the programs and corresponding open source code freely available under the MIT license at SourceForge, as requested.